# Calibrating the Rigged Lottery: Making All Tickets Reliable

**Bowen Lei**
Texas A&M University
`bowenlei@stat.tamu.edu`

**Ruqi Zhang**
Purdue University
`ruqiz@purdue.edu`

**Dongkuan Xu**
North Carolina State University
`dxu27@ncsu.edu`

**Bani Mallick**
Texas A&M University
`bmallick@stat.tamu.edu`

## Abstract

Although sparse training has been successfully used in various resource-limited deep learning tasks to save memory, accelerate training, and reduce inference time, the reliability of the produced sparse models remains unexplored. Previous research has shown that deep neural networks tend to be over-confident, and we find that sparse training exacerbates this problem. Therefore, calibrating the sparse models is crucial for reliable prediction and decision-making. In this paper, we propose a new sparse training method to produce sparse models with improved confidence calibration. In contrast to previous research that uses only one mask to control the sparse topology, our method utilizes two masks, including a deterministic mask and a random mask. The former efficiently searches and activates important weights by exploiting the magnitude of weights and gradients. While the latter brings better exploration and finds more appropriate weight values by random updates. Theoretically, we prove our method can be viewed as a hierarchical variational approximation of a probabilistic deep Gaussian process. Extensive experiments on multiple datasets, model architectures, and sparsities show that our method reduces ECE values by up to 47.8% and simultaneously maintains or even improves accuracy with only a slight increase in computation and storage burden.

## 1 Introduction

Sparse training is gaining increasing attention and has been used in various deep neural network (DNN) learning tasks (Evci et al., 2020; Dietrich et al., 2021; Bibikar et al., 2022). In sparse training, a certain percentage of connections are maintained being removed to save memory, accelerate training, and reduce inference time, enabling DNNs for resource-constrained situations. The sparse topology is usually controlled by a mask, and various sparse training methods have been proposed to find a suitable mask to achieve comparable or even higher accuracy compared to dense training (Evci et al., 2020; Liu et al., 2021; Schwarz et al., 2021). However, in order to deploy the sparse models in real-world applications, a key question remains to be answered: how reliable are these models?

There has been a line of work on studying the reliability of dense DNNs, which means that DNNs should know what it does not know (Guo et al., 2017; Nixon et al., 2019; Wang et al., 2021). In other words, a model's confidence (the probability associated with the predicted class label) should reflect its ground truth correctness likelihood. A widely used reliability metric is Expected Calibration Error (ECE) (Guo et al., 2017), which measures the difference between confidence and accuracy, with a lower ECE indicating higher reliability. However, prior research has shown that DNNs tend to be over-confident (Guo et al., 2017; Rahaman et al., 2021; Patel et al., 2022), suggesting DNNs may be too confident to notice incorrect decisions, leading to safety issues in real-world applications, e.g., automated healthcare and self-driving cars (Jiang et al., 2012; Bojarski et al., 2016).

In this work, we for the *first* time identify and study the reliability problem of sparse training. We start with the question of how reliable the current sparse training is. We find that the over-confidence

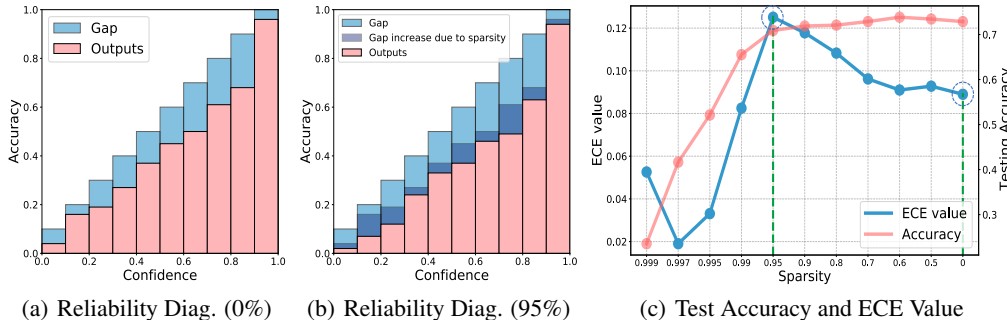

(a) Reliability Diag. (0%)    (b) Reliability Diag. (95%)    (c) Test Accuracy and ECE Value

Figure 1: Reliability diagrams for (a) the dense model and (b) the sparse model. The sparse model is more over-confident than the dense model. (c) the scatter plot of test accuracy (%) and ECE value at different sparsities. From the high sparse model to the dense model, the ECE value first decreases, then increases, and then decreases again, showing a double descent pattern.

problem becomes even more pronounced when sparse training is applied to ResNet-50 on CIFAR-100. Figures 1 (a)-(b) show that the gap (blue area) between confidence and accuracy of the sparse model (95% sparsity) is larger than that of the dense model (0% sparsity), implying the sparse model is more over-confident than the dense model. Figure 1 (c) shows the test accuracy (pink curve) and ECE value (blue curve, a measure of reliability) (Guo et al., 2017) at different sparsities. When the accuracy is comparable to dense training (0%-95%), we observe that the ECE values increase with sparsity, implying that the problem of over-confidence becomes more severe at higher sparsity. And when the accuracy decreases sharply (>95%), the ECE value first decreases and then increases again. This leads to a *double descent* phenomenon (Nakkiran et al., 2021) when we view the ECE value curve from left to right (99.9%-0%) (see Section 6 for more discussion).

To improve the reliability, we propose a new sparse training method to produce well-calibrated predictions while maintaining a high accuracy performance. We call our method "The Calibrated Rigged Lottery" or CigL. Unlike previous sparse training methods with only one mask, our method employs two masks, including a deterministic mask and a random mask, to better explore the sparse topology and weight space. The deterministic one efficiently searches and activates important weights by exploiting the magnitude of weights/gradients. And the random one, inspired by dropout, adds more exploration and leads to better convergence. When near the end of training, we collect weights & masks at each epoch, and use the designed weight & mask averaging procedure to obtain one sparse model. From theoretical analysis, we show our method can be viewed as a hierarchical variational approximation (Ranganath et al., 2016) to a probabilistic deep Gaussian process (Gal & Ghahramani, 2016), which leads to a large family of variational distributions and better Bayesian posterior approximations. Our contributions are summarized as follows:

- We for the first time identify and study the reliability problem of sparse training and find that sparse training exacerbates the over-confidence problem of DNNs.

- We then propose CigL, a new sparse training method that improves confidence calibration with comparable and even higher accuracy.

- We prove that CigL can be viewed as a hierarchical variational approximation to a probabilistic deep Gaussian process which improves the calibration by better characterizing the posterior.

- We perform extensive experiments on multiple benchmark datasets, model architectures, and sparsities. CigL reduces ECE values by up to **47.8%** and simultaneously maintain or even improve accuracy with only a slight increase in computational and storage burden.

## 2 RELATED WORK

### 2.1 SPARSE TRAINING

As the scale of models continues to grow, there is an increasing attention to the sparse training which maintains sparse weights throughout the training process. Different sparse training methods have been investigated, and various pruning and growth criteria, such as weight/gradient magnitude,

are designed (Mocanu et al., 2018; Bellec et al., 2018; Frankle & Carbin, 2019; Mostafa & Wang, 2019; Dettmers & Zettlemoyer, 2019; Evci et al., 2020; Jayakumar et al., 2020; Liu et al., 2021; Özdenizci & Legenstein, 2021; Zhou et al., 2021; Schwarz et al., 2021; Yin et al., 2022). However, sparse training is more challenging in the weight space exploration because sparse constraints cut off update routes and produce spurious local minima (Evci et al., 2019; Sun & Li, 2021; He et al., 2022). There are some studies that have started to promote exploration, while they primarily pursue high accuracy and might add additional costs (Liu et al., 2021; Huang et al., 2022). Most sparse training methods use only one mask to determine the sparse topology, which is insufficient for achieving adequate exploration, and existing multi-mask methods are not designed for improved exploration (Xia et al., 2022; Bibikar et al., 2022) (more details in Section D.4).

## 2.2 Confidence Calibration in DNNs

Many studies have investigated whether the confidences of DNNs are well-calibrated (Guo et al., 2017; Nixon et al., 2019; Zhang et al., 2020), and existing research has found DNNs tend to be over-confident (Guo et al., 2017; Rahaman et al., 2021; Patel et al., 2022), which may mislead our choices and cause unreliable decisions in real-world applications. To improve confidence calibration, a widely-used method is temperature scaling (Guo et al., 2017), which adds a scaling parameter to the softmax formulation and adjusts it on a validation set. Some other works incorporate regularization in the training, such as Mixup (Zhang et al., 2017) and label smoothing (Szegedy et al., 2016). In addition, Bayesian methods also show the ability to improve calibration, such as Monte Carlo Dropout (Gal & Ghahramani, 2016) and Bayesian deep ensembles (Ashukha et al., 2020). However, they mainly focus on dense training. Studies have been conducted on reliability of sparse DNNs (more details in Section D.5), but they target on pruning, starting with a dense model to gradually increase sparsity, which reduces the exploration challenge (Venkatesh et al., 2020; Chen et al., 2022). They still find that uncertainty measures are more sensitive to pruning than generalization metrics, indicating the sensitivity of reliability to sparsity. Yin et al. (2022) studies sparse training, but it aims to boost the performance and brings limited improvement in reliability. Therefore, how to obtain a well-calibrated DNN in sparse training is more challenging and remains unknown.

## 3 Method

We propose a new sparse training method, CigL, to improve the confidence calibration of the produced sparse models, which simultaneously maintains comparable or even higher accuracy. Specifically, CigL starts with a random sparse network and uses two masks to control the sparse topology and explore the weight space, including a deterministic mask and a random mask. The former is updated periodically to determine the non-zero weights, while the latter is sampled randomly in each iteration to bring better exploration in the model update. Then, with the designed weight & mask averaging, we combine information about different aspects of the weight space to obtain a single output sparse model. Our CigL method is outlined in Algorithm 1.

### 3.1 Deterministic Mask & Random Mask

In our CigL, we propose to utilize two masks, a deterministic mask $M$ and a random mask $Z$, to search for a sparse model with improved confidence calibration and SOTA accuracy. We will first describe the two masks in detail and discuss how to set their sparsity.

**The deterministic mask** controls the entire sparse topology with the aim of finding a well-performing sparse model. That is, the mask determines which weights should be activated and which should not. Inspired by the widely-used sparse training method RigL (Evci et al., 2020), we believe a larger weight/gradient magnitude implies that the weight is more helpful for loss reduction and needs to be activated. Thus, CigL removes a portion of the weights with small magnitudes, and activates new weights with large gradient magnitudes at fixed time intervals $\Delta T$.

**The random mask** allows the model to better explore the weight space under sparsity constraints. In each iteration prior to backpropagation, the mask is randomly drawn from Bernoulli distribution. In this way, the mask randomly selects a portion of the non-zero weights to be temporarily deactivated and forces the model to explore more in other directions of the weight space, which adds more randomness in the weight update step and leads to a better exploration of the weight space compared

to one mask strategy. As a result, the model is more likely to jump out of spurious local minima while avoiding deviations from the sparse topology found by the deterministic mask.

**The sparsity setting** of the two masks is illustrated as below. On the one hand, the deterministic mask is responsible for the overall sparsity of the output sparse model. Suppose we want to train a network with 95% sparsity, the deterministic mask will also have the same sparsity, with 5% of the elements being 1. On the other hand, the random mask deactivates some non-zero weights during the training process, producing temporary models with increasing sparsity. Since highly sparse models (like 95% sparsity) are sensitive to further increases in sparsity, we set a low sparsity, such as 10%, for the random mask so that no significant increases in sparsity and no dramatic degradation in performance occurs in these temporary models.

### 3.2 Weight & Mask Averaging

With the two masks designed above, we propose a weight & mask averaging procedure to obtain one single sparse model with improved confidence calibration and comparable or even higher accuracy. We formalize this procedure as follows. We first iteratively update the two

---

**Algorithm 1** CigL

**Input:** initialize $\boldsymbol{W}^{(0)}$, $\boldsymbol{M}$, and $\boldsymbol{W}_{\text{CigL}} = \text{None}$, set epoch length $m$, update interval $\Delta T$, number of iterations $T$, start iteration for weight & mask averaging $T^*$, random mask rate $p$, and learning rate $\alpha_t$
**for** $t = 1$ **to** $T$ **do**
  Sample a mini-batch data $\boldsymbol{B}_t$ with size $n$
  **if** $t \mod \Delta T = 0$ **then**
    Update mask $\boldsymbol{M}$ using weights and gradients
    Prune and regrow weights $\boldsymbol{W}^{(t)}$ based on $\boldsymbol{M}$
  **end if**
  Sample mask $\boldsymbol{Z}^{(t)}$ and $\boldsymbol{Z}_{ij}^{(t)} = \text{Bernoulli}(p)$
  Update sparse weights: $\boldsymbol{W}^{(t)} = \boldsymbol{W}^{(t-1)} - \alpha_t \boldsymbol{M} \odot \boldsymbol{Z}^{(t)} \odot \nabla L(\boldsymbol{M} \odot \boldsymbol{Z}^{(t)} \odot \boldsymbol{W}^{(t-1)}; \boldsymbol{B}_t)$
  **if** $t \mod m = 0$ and $t > T^*$ **then**
    **if** $\boldsymbol{W}_{\text{CigL}} = \text{None}$ **then**
      $\boldsymbol{W}_{\text{CigL}} = \boldsymbol{M} \odot \boldsymbol{Z}^{(t)} \odot \boldsymbol{W}^{(t)}$
      $n_{\text{models}} = 1$
    **else**
      $\boldsymbol{W}_{\text{CigL}} = \frac{\boldsymbol{W}_{\text{CigL}} \cdot n_{\text{models}} + \boldsymbol{M} \odot \boldsymbol{Z}^{(t)} \odot \boldsymbol{W}^{(t)}}{n_{\text{models}} + 1}$
      $n_{\text{models}} = n_{\text{models}} + 1$
    **end if**
  **end if**
**end for**
**Output:** Sparse Model Weights $\boldsymbol{W}_{\text{CigL}}$

---

masks and model weights. Consistent with widely used sparse training methods (Evci et al., 2020; Liu et al., 2021), the deterministic mask stops updating near the end of the training process. While we still continuously draw different random masks from the Bernoulli distribution and collect a pair of sparse weights and random masks $\{\boldsymbol{Z}^{(t)}, \boldsymbol{W}^{(t)}\}$ at each epoch after $T^*$-th epoch. Then, we can produce multiple temporary sparse models $\boldsymbol{Z}^{(t)} \odot \boldsymbol{W}^{(t)}$ with different weight values and different sparse topologies, which contain more knowledge about the weight space than the single-mask training methods. Finally, inspired by a popular way of combining models (Izmailov et al., 2018; Wortsman et al., 2022), we obtain the single output sparse model by averaging the weights of these temporary sparse models, which can be viewed as a mask-based weighted averaging.

## 4 Making All Tickets Reliable

### 4.1 CigL with Better Confidence Calibration

Obtaining reliable DNNs is more challenging in sparse training, and we will show why our CigL provides a solution to this problem. Bayesian methods have shown the ability to improve confidence calibration (Gal & Ghahramani, 2016; Ashukha et al., 2020), but they become more difficult to fit the posterior well under sparse constraints, limiting their ability to solve unreliable problems. We find that CigL can be viewed as a hierarchical Bayesian method (shown in Section 4.2), which improves confidence calibration by performing better posterior approximations in two ways discussed below.

On the one hand, the model is more challenging to fully explore the weight space due to sparsity constraint, and inappropriate weight values can also negatively affect the mask search. During sparse training, a large percentage of connections are removed, cutting off the update routes and thus narrowing the family of Bayesian proposal distributions. This leads to more difficult optimization and sampling. To overcome this issue, our CigL adds a hierarchical structure to the variational distributions so that we have a larger family of distributions, allowing it to capture more complex marginal distributions and reducing the difficulty of fitting the posterior.

On the other hand, when sparsity constraints are added, the posterior landscape changes, leading to more complex posterior distributions. One example is the stronger correlation between hidden

variables, such as the random mask $\boldsymbol{Z}$ and weight $\boldsymbol{W}$ (shown in the Appendix C.1). In a dense model, the accuracy does not change much if we randomly draw $\boldsymbol{Z}$ and use $\boldsymbol{Z} \odot \boldsymbol{W}$ compared to using $\boldsymbol{W}$. However, in high sparsity like 95%, we see a significant accuracy drop when $\boldsymbol{Z} \odot \boldsymbol{W}$ is used compared to using $\boldsymbol{W}$. Thus, in CigL, the pairings of $\boldsymbol{Z}$ and $\boldsymbol{W}$ are collected to capture the correlation, leading to a better posterior approximation.

## 4.2 CigL as a Hierarchical Bayesian Approximation

We prove that training sparse neural networks with our CigL are mathematically equivalent to approximating the probabilistic deep GP (Damianou & Lawrence, 2013; Gal & Ghahramani, 2016) with hierarchical variational inference. We show that the objective of CigL is actually to minimize the Kullback–Leibler (KL) divergence between a hierarchical variational distribution and the posterior of a deep GP. During our study, we do not restrict the architecture, allowing the results applicable to a wide range of applications. Detailed derivation is shown in Appendix B.

We first present the minimisation objective function of CigL for a sparse neural network (NN) model with $L$ layers and loss function $E$. The sparse weights and bias of the $l$-th layer are denoted by $\boldsymbol{W}_l \in \mathbb{R}^{K_i \times K_{i-1}}$ and $\boldsymbol{b}_l \in \mathbb{R}^{K_i}$ ($l = 1, \cdots, L$), and the output prediction is denoted by $\widehat{\boldsymbol{y}}_i$. Given data $\{\boldsymbol{x}_i, y_i\}$, we train the NN model by iteratively update the deterministic mask and the sparse weights. Since the random mask is drawn from a Bernoulli distribution, it has no parameters that need to be updated. For deterministic mask updates, we prune weights with the smaller weight magnitude and regrow weights with larger gradient magnitude. For the weight update, we minimise Eq. (1) which is composed of the difference between $\widehat{\boldsymbol{y}}_i$ and the true label $\boldsymbol{y}_i$ and a $L_2$ regularisation.

$$\mathcal{L}_{\text{CigL}} := \frac{1}{N} \sum_{i=1}^{N} E(y_i, \widehat{y}_i) + \lambda \sum_{l=1}^{L} (||\boldsymbol{W}_l||_2^2 + ||\boldsymbol{b}_l||_2^2). \tag{1}$$

Then, we derive the minimization objective function of Deep GP which is a flexible probabilistic NN model that can model the distribution of functions (Gal & Ghahramani, 2016). Taking regression as an example, we assume that $\boldsymbol{W}_l$ is a random matrix and $\boldsymbol{w} = \{\boldsymbol{W}_l\}_{l=1}^{L}$, and denote the prior by $p(\boldsymbol{w})$. Then, the predictive distribution of the deep GP can be expressed as Eq. (2) where $\tau > 0$

$$p(\boldsymbol{y}|\boldsymbol{x}, \boldsymbol{X}, \boldsymbol{Y}) = \int p(\boldsymbol{y}|\boldsymbol{x}, \boldsymbol{w}) p(\boldsymbol{w}|\boldsymbol{X}, \boldsymbol{Y}) d\boldsymbol{w}, \tag{2}$$

$$p(\boldsymbol{y}|\boldsymbol{x}, \boldsymbol{w}) = \mathcal{N}(\boldsymbol{y}; \widehat{\boldsymbol{y}}, \tau^{-1}\boldsymbol{I}), \quad \widehat{\boldsymbol{y}} = \sqrt{\frac{1}{K_L}} \boldsymbol{W}_L \sigma \left( \cdots \sqrt{\frac{1}{K_1}} \boldsymbol{W}_2 \sigma \left( \boldsymbol{W}_1 \boldsymbol{x} + \boldsymbol{u}_1 \right) \right).$$

The posterior distribution $p(\boldsymbol{w}|\boldsymbol{X}, \boldsymbol{Y})$ is intractable, and one way of training the deep GP is variational inference where a family of tractable distributions $q(\boldsymbol{w})$ is chosen to approximate the posterior. Specifically, we define the hierarchy of $q(\boldsymbol{w})$ as Eq. (3):

$$q(\boldsymbol{W}_{lij}|\boldsymbol{Z}_{lij}, \boldsymbol{U}_{lij}, \boldsymbol{M}_l) \sim \boldsymbol{Z}_{lij} \cdot \mathcal{N}(\boldsymbol{M}_{lij}\boldsymbol{U}_{lij}, \sigma^2) + (1 - \boldsymbol{Z}_{lij}) \cdot \mathcal{N}(0, \sigma^2),$$

$$q(\boldsymbol{M}_l|\boldsymbol{U}_l) \propto \exp(\boldsymbol{M}_l \odot (|\boldsymbol{U}_l| + |\nabla\boldsymbol{U}_l|)), \quad \boldsymbol{U}_{lij} \sim \mathcal{N}(\boldsymbol{V}_{lij}, \sigma^2), \quad \boldsymbol{Z}_{lij} \sim \text{Bernoulli}(p_l), \tag{3}$$

where $l$, $i$ and $j$ denote the layer, row, and column index, $\boldsymbol{M}_l$ is a matrix with 0's and constrained 1's, $\boldsymbol{W}_l$ is the sparse weights, $\boldsymbol{U}_l$ is the variational parameters, and $\boldsymbol{V}_l$ is the variational hyper parameters.

Then, we iteratively update $\boldsymbol{M}_l$ and $\boldsymbol{W}_l$ to approximate the posterior. For the update of $\boldsymbol{M}_l$, we obtain a point estimate by maximising $q(\boldsymbol{M}_l|\boldsymbol{U}_l)$ under the sparsity constraint. In pruning step, since the gradient magnitudes $|\nabla\boldsymbol{U}_l|$ can be relatively small compared to the weight magnitudes $|\boldsymbol{U}_l|$ after training, we can use $\exp(\boldsymbol{M}_l \odot |\boldsymbol{U}_l|)$ to approximate the distribution. In regrowth step, since the inactive weights are zero, we directly compare the gradient magnitudes $\exp(\boldsymbol{M}_l \odot |\nabla\boldsymbol{U}_l|)$. Thus, the update of $\boldsymbol{M}_l$ is aligned with the update in CigL.

For $\boldsymbol{W}_l$, we minimise the KL divergence between $q(\boldsymbol{w})$ and the posterior of deep GP as Eq. (4)

$$- \int q(\boldsymbol{w}) \log p(\boldsymbol{Y}|\boldsymbol{X}, \boldsymbol{w}) d\boldsymbol{w} + D_{\text{KL}}(q(\boldsymbol{w}) \| p(\boldsymbol{w})). \tag{4}$$

For the first term in Eq. (4), we can first rewrite it as $-\sum_{n=1}^{N} \int q(\boldsymbol{w}) \log p(y_n|\boldsymbol{x}_n, \boldsymbol{w})$. Then, we can approximate each integration in the sum with a single estimate $\widehat{\boldsymbol{w}}$. For the second term in Eq. (4),

we can approximate it as $\sum_{l=1}^{L}(\frac{p_l}{2}\|U_l\|_2^2 + \frac{1}{2}\|u_l\|_2^2)$. As a result, we can derive the objective as

$$\mathcal{L}_{\text{GP}} := \frac{1}{N}\sum_{i=1}^{N}\frac{-\log p(y_n|x_n,\widehat{w})}{\tau} + \sum_{l=1}^{L}(\frac{p_l}{2}\|U_l\|_2^2 + \frac{1}{2}\|u_l\|_2^2), \tag{5}$$

which is shown to have the same form as the objective in Eq. (1) with appropriate hyperparameters for the deep GP. Thus, the update of $W_l$ is also consistent with the update in CigL. This suggests that our CigL can be viewed as an approximation to the deep GP using hierarchical variational inference. The final weight & mask averaging procedure can be incorporated into the Bayesian paradigm as an approximation to the posterior distribution (Srivastava et al., 2014; Maddox et al., 2019).

### 4.3 Connection to Dropout

Our CigL can be seen as a new version of Dropout, and our random mask $Z$ is related to the Dropout mask. Dropout is a widely used method to overcome the overfitting problem (Hinton et al., 2012; Wan et al., 2013; Srivastava et al., 2014). Two widely used types are unit dropout and weight dropout, which randomly discard units (neurons) and individual weights at each training step, respectively. Both methods use dropouts only in the training phase and remove them in the testing phase, which is equivalent to discarding $Z$ and only using $W$ for prediction. However, simply dropping $Z$ can be detrimental to the fit of the posterior. Thus, MC dropout collects multiple models by randomly selecting multiple dropout masks, which is equivalent to extracting multiple $Z$ and using one $W$ for prediction. However, only using one $W$ neither fully expresses the posterior landscape nor captures the correlation between $Z$ and $W$. In contrast, our CigL uses multiple pairings of $Z$ and $W$, which can better approximate the posterior under sparsity constraints.

### 4.4 Connection to Weight Averaging

Our weight & mask averaging can be seen as an extension of weight averaging (WA), which averages the weights of multiple model samples to produce a single output model (Izmailov et al., 2018; Wortsman et al., 2022). Compared to deep ensembles (Ashukha et al., 2020), WA outputs only one model, which reduces the forward FLOPs and speeds up prediction. When these model samples are located in one low error basin, it usually leads to wider optima and better generalization. However, although WA can produce better generalization, it does not improve the confidence calibration (Wortsman et al., 2022). In contrast to WA, our weight & mask averaging uses masks for weighted averaging and improves the confidence calibration with similar FLOPs in the prediction.

## 5 Experiments

We perform a comprehensive empirical evaluation of CigL, comparing it with the popular baseline method RigL (Evci et al., 2020). RigL is a popular sparse training method that uses weights magnitudes to prune and gradient magnitudes to grow connections.

**Datasets & Model Architectures:** We follow the settings in Evci et al. (2020) for a comprehensive comparison. Our experiments are based on three benchmark datasets: CIFAR-10 and CIFAR-100 (Krizhevsky et al., 2009) and ImageNet-2012 (Russakovsky et al., 2015). For model architectures, we used ResNet-50 (He et al., 2016) and Wide-ResNet-22-2 (Zagoruyko & Komodakis, 2016). We repeat all experiments 3 times and report the mean and standard deviation.

**Sparse Training Settings:** For sparse training, we check multiple sparsities, including 80%, 90%, 95%, and 99%, which can sufficiently reduce the memory requirement and is of more interest.

**Implementations:** We follow the settings in (Evci et al., 2020; Sundar & Dwaraknath, 2021). The parameters are optimized by SGD with momentum. For the learning rate, we use piecewise constant decay scheduler. For CIFAR-10 and CIFAR-100, we train all the models for 250 epochs with a batch size of 128. For ImageNet, we train all the models for 100 epochs with a batch size of 64.

### 5.1 Comparison between Popular Sparse Training Method

**Results on CIFAR-10 and CIFAR-100.** We first compare our CigL and RigL by the expected calibration error (ECE) (Guo et al., 2017), a popular measure of the discrepancy between a model's

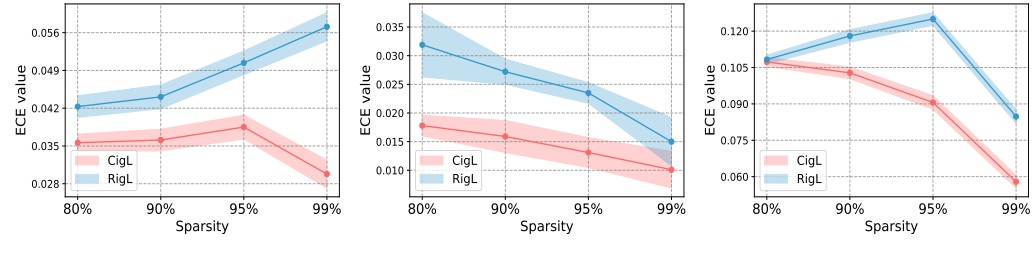

(a) CIFAR-10, ResNet-50     (b) CIFAR-10, Wide-ResNet-22-2     (c) CIFAR-100, ResNet-50

Figure 2: ECE value comparison between CigL and RigL at different sparsities (80%, 90%, 95%, 99%). Compared to RigL, CigL produces sparse models with smaller ECE values.

Table 1: Testing accuracy (%) comparison between CigL and RigL at different sparsities (80%, 90%, 95%, 99%). Compared to RigL, CigL maintains comparable or higher test accuracy.

|  | CIFAR-10 | | CIFAR-100 | |
| --- | --- | --- | --- | --- |
|  | RIGL | CIGL | RIGL | CIGL |
| 80% SPARSITY | 94.02 (0.115) | **94.75** (0.107) | 72.08 (0.109) | **76.84** (0.089) |
| 90% SPARSITY | 93.84 (0.184) | **94.56** (0.189) | 71.90 (0.172) | **76.24** (0.181) |
| 95% SPARSITY | 93.19 (0.198) | **94.20** (0.202) | 70.90 (0.210) | **74.71** (0.197) |
| 99% SPARSITY | 91.31 (0.205) | **92.42** (0.196) | 65.57 (0.208) | **66.42** (0.206) |

confidence and true accuracy, with a lower ECE indicating better confidence calibration and higher reliability. In Figure 2, the pink and blue curves represent CigL and RigL, respectively, where the colored ares represent the 95% confidence intervals. We can see that the pink curves are usually lower than the blue curves for different sparsities (80%, 90%, 95%, 99%), which implies that our CigL can reduce the ECE and improve the confidence calibration of the produced sparse models.

Apart from ECE value, we also compare our CigL and RigL by the testing accuracy for multiple sparsities (80%, 90%, 95%, 99%). We summarize the results for sparse ResNet-50 in Table 1. It is observed that CigL tends to bring comparable or higher accuracy, which demonstrates that CigL can simultaneously maintain or improve the accuracy.

**Results on ImageNet-2012**. We also compare the ECE values and test accuracy of our CigL and RigL on a larger dataset, ImageNet-2012, where the sparsity of ResNet-50 is 80% and 90%. As shown in Figure 3, the pink and blue bars represent our CigL and RigL, respectively. For the comparison of ECE values in (a), the pink bars are shorter than the blue bars, indicating an improved reliability of the sparse model produced by CigL. For the test accuracy comparison in (b), the pink and blue bars are very similar in height, implying that the accuracy of CigL is comparable to that of RigL.

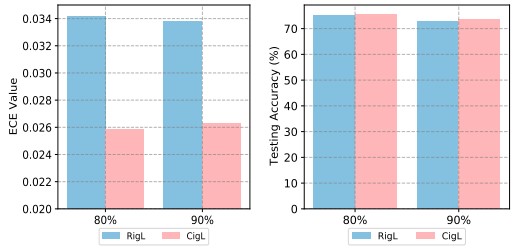

(a) ECE value (RM)     (b) Test accuracy (RM)

Figure 3: ECE value and test accuracy(%) of CigL and RigL at 80% & 90% sparsities on ImageNet-2012. Compared with RigL, CigL has smaller ECE values and comparable test accuracies.

## 5.2 COMPARISON BETWEEN DIFFERENT DROPOUT METHODS

In this section, since our CigL is related to dropout methods, we compare our CigL with RigL using existing popular dropout methods, namely weight dropout (W-DP) and MC dropout (MC-DP). **The comparison of test accuracy** is shown in Table 2. Our CigL usually provides a comparable or higher accuracy compared to RigL. However, using weight dropout and MC dropout in RigL usually result in a decrease in accuracy. We also summarize **the comparison of the ECE value** between CigL and different dropout methods in Table 3. For each sparsity and architecture, we have marked in bold those cases where the ECE value is significantly reduced ($\geq 15\%$ reduction compared to RigL). Our CigL are always bolded, indicating its ability to reduce ECE value and increase reliability in sparse training. But RigL + weight dropout does not significantly reduce ECE values in almost all cases and RigL + MC dropout also does not improve the calibration in highly sparse cases (99% sparsity).

Table 2: Testing accuracy (%) comparison between CigL, RigL + weight dropout (W-DP), and RigL + MC dropout (MC-DP) at different sparsities (80%, 90%, 95%, 99%). Compared to RigL, RigL + W-DP, and RigL + MC-DP, CigL maintains comparable or higher test accuracy.

|  |  | 80% SPARSITY | 90% SPARSITY | 95% SPARSITY | 99% SPARSITY |
|---|---|---|---|---|---|
| RESNET-50 | RIGL | 94.02 (0.115) | 93.84 (0.184) | 93.19 (0.198) | 91.31 (0.205) |
|  | RIGL + W-DP | 93.26 (0.114) | 93.47 (0.186) | 92.71 (0.193) | 89.99 (0.210) |
|  | RIGL + MC-DP | 93.39 (0.105) | 93.71 (0.181) | 92.87 (0.205) | 89.84 (0.212) |
|  | CIGL | **94.75** (0.107) | **94.56** (0.189) | **94.20** (0.202) | **92.42** (0.196) |
| WRN-22-2 | RIGL | 93.12 (0.188) | 92.26 (0.187) | 91.02 (0.179) | 83.82 (0.224) |
|  | RIGL + W-DP | 91.77 (0.182) | 91.44 (0.191) | 89.66 (0.183) | 80.42 (0.215) |
|  | RIGL + MC-DP | 91.75 (0.149) | 91.49 (0.187) | 89.39 (0.177) | 77.48 (0.198) |
|  | CIGL | **93.95** (0.088) | **93.05** (0.219) | **91.34** (0.171) | **83.96** (0.189) |

Table 3: Testing ECE comparison between CigL, RigL + weight dropout (W-DP), and RigL + MC dropout (MC-DP) at different sparsities (80%, 90%, 95%, 99%). Compared to RigL + W-DP and RigL + MC-DP, CigL more consistently achieves a significant reduction in the ECE value of RigL.

|  |  | 80% SPARSITY | 90% SPARSITY | 95% SPARSITY | 99% SPARSITY |
|---|---|---|---|---|---|
| RESNET-50 | RIGL | 0.0423 (0.001) | 0.0441 (0.001) | 0.0504 (0.001) | 0.0571 (0.001) |
|  | RIGL + W-DP | 0.0504 (0.002) | 0.0438 (0.001) | 0.0462 (0.002) | **0.0315** (0.002) |
|  | RIGL + MC-DP | **0.0322** (0.001) | **0.0200** (0.001) | **0.0121** (0.001) | 0.0528 (0.002) |
|  | CIGL | **0.0356** (0.001) | **0.0361** (0.001) | **0.0385** (0.001) | **0.0298** (0.001) |
| WRN-22-2 | RIGLT | 0.0319 (0.003) | 0.0272 (0.001) | 0.0235 (0.001) | 0.0150 (0.002) |
|  | RIGL + W-DP | 0.0433 (0.003) | 0.0348 (0.002) | 0.0256 (0.002) | 0.0174 (0.003) |
|  | RIGL + MC-DP | **0.0159** (0.001) | **0.0077** (0.002) | 0.0384 (0.001) | 0.1502 (0.002) |
|  | CIGL | **0.0178** (0.001) | **0.0159** (0.001) | **0.0131** (0.001) | **0.0101** (0.002) |

## 5.3 COMPARISON BETWEEN OTHER CALIBRATION METHODS

In this section, we compare our CigL with existing popular calibration methods, including mixup (Zhang et al., 2017), temperature scaling (TS) (Guo et al., 2017), and label smoothing (LS) (Szegedy et al., 2016). **The testing ECE are depicted in Figure 4, where the pink and blue polygons represent CigL and other calibration methods, respectively. We can see that CigL usually gives smaller polygons, indicating a better confidence calibration.

## 5.4 ABLATION STUDIES

We do ablation studies to demonstrate the importance of each component in our CigL, where we train sparse networks using our CigL without random masks (CigL w/o RM) and CigL without weight & mask averaging (CigL w/o WMA), respectively. In CigL w/o RM, we search for sparse topologies using only the deterministic mask. In CigL w/o WMA, we collect multiple model samples and use prediction averaging during testing. Figures 5(a)-(b) show **the effect of random masks** on the test accuracy and ECE values, where the blue, green, and pink bars represent RigL, CigL w/o RM, and CigL, respectively. We can see that if we remove the random mask, we can still obtain an improvement in accuracy compared to RigL. However, the ECE values do not decrease as much as CigL, indicating that the CigL w/o RM is not as effective as CigL in improving the confidence calibration. Figures 5(c)-(d) further show **the effect of weight & mask averaging**. We can see that without using weight & mask averaging, the accuracy decreases and the ECE value increases in high sparsity such as 95% and 99%, demonstrating the importance of weight & mask averaging.

## 6 DISCUSSION & CONCLUSION

We for the first time identify and study the reliability problem of sparse training and find that sparse training exacerbates the over-confidence problem of DNNs. We then develop a new sparse training

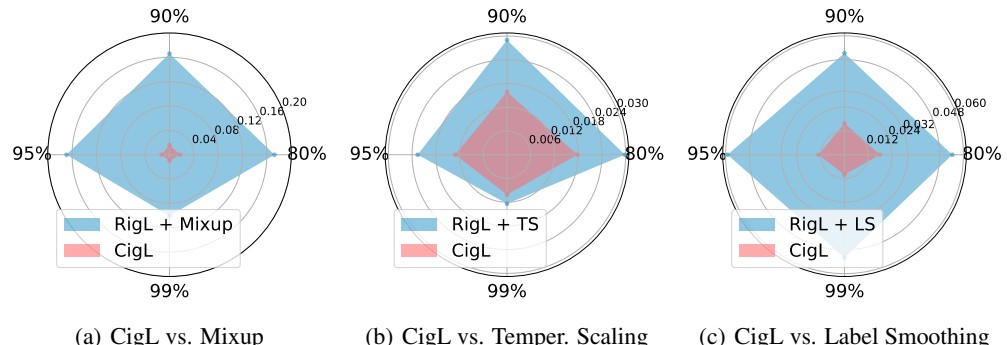

(a) CigL vs. Mixup     (b) CigL vs. Temper. Scaling     (c) CigL vs. Label Smoothing

Figure 4: ECE value comparison between CigL and RigL + other calibration methods at different sparsities (80%, 90%, 95%, 99%). The pink polygons (CigL) are smaller than the blue polygons (other calibration methods), indicating a better confidence calibration using CigL compared to (a) Mixup, (b) Temperature scaling, and (c) Label smoothing.

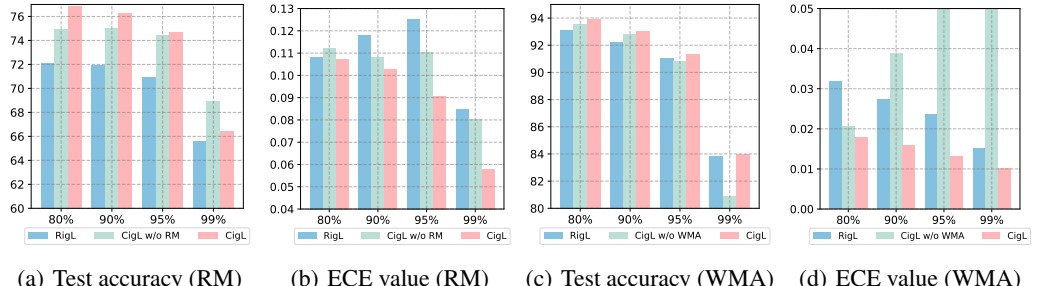

(a) Test accuracy (RM)    (b) ECE value (RM)    (c) Test accuracy (WMA)    (d) ECE value (WMA)

Figure 5: Ablation studies: test accuracy(%) and ECE value comparison between CigL, CigL without random mask (CigL w/o RM), and CigL without weight & mask averaging (CigL w/o WMA) at different sparsities (80%, 90%, 95%, 99%). Compared to (a)-(b) CigL w/o RM and (c)-(d) CigL w/o WMA, CigL more consistently produces sparse models with low ECE values and high accuracy.

method, CigL, to produce reliable sparse models, which can simultaneously maintain or even improve accuracy with only a slight increase in computational and storage burden. Our CigL utilizes two masks, including a deterministic mask and a random mask, which allows the sparse model to better explore the weight space. Then, we design weight & mask averaging method to combine multiple sparse weights and random masks into a single model with improved reliability. We prove that CigL can be viewed as a hierarchical variational approximation to the probabilistic deep Gaussian process. Experiments results on multiple benchmark datasets, model architectures, and sparsities show that our CigL reduces ECE values by up to **47.8%** with comparable or higher accuracy.

One phenomenon we find worth discussing is the *double descent* in reliability of sparse training. Nakkiran et al. (2021) first observed this double descent phenomenon in DNNs, where as the model size, data size, or training time increases, the performance of the model first improves, then gets worse, and then improves again. Consistent with the previous definition, we consider sparsity and reliability as the measures of model size and performance, respectively. Then, as shown in the Figure 1 (c), as the sparsity decreases (model size increases), the reliability (model performance) gets better, then gets worse, and then gets better again. To explain this phenomenon, we divided sparsity into four phases, from the left (99.9%) to the right (0%), by drawing an analogy between the phases and model accuracy and size. **(a) The sparse model starts as a poor model**, which is too sparse to learn the data well (low reliability & accuracy). **(b) It gradually becomes equivalent to a shallow model** that can learn some patterns but is not flexible enough to learn all the data well (high reliability & moderate level of accuracy). **(c) Then, it moves to a sparse deep model** that can accommodate complex patterns but suffers from poor exploration (low reliability & high accuracy). **(d) Finally, it reaches a dense deep model** with over-confidence issues (moderate level of reliability & high accuracy). It is observed that at around 95% sparsity, the sparse model can achieve comparable accuracy and high sparsity at the same time, which makes it important in practical applications. However, the ECE value is at the peak of the double-descent curve at this point, which implies that the reliability of the sparse model is at a low level. Thus, our CigL smooths the double descent curve and produce reliable models on those important high sparsity levels.

## ACKNOWLEDGMENTS

This research was partially supported by NSF Grant No. NSF CCF-1934904 (TRIPODS).

## REPRODUCIBILITY STATEMENT

The implementation code can be found in https://github.com/StevenBoys/CigL. All datasets and code platform (PyTorch) we use are public.

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

# A  APPENDIX: BACKGROUND

In this section, we briefly summarize CigL, Gaussian processes, and hierarchical variational inference, which will be used to support the main theoretical analysis of this work.

## A.1  SPARSE TRAINING: CIGL

We first review our CigL method for the case of a single hidden layer neural network (NN). This is done for ease of notation, and it is straightforward to generalise to multiple layers (Gal & Ghahramani, 2016). Denote by $\boldsymbol{W}_1$, $\boldsymbol{W}_2$ the sparse weight matrices connecting the first layer to the hidden layer and connecting the hidden layer to the output layer respectively. For the sparse mask controlling the sparse topology, we use $\boldsymbol{M}_1$, $\boldsymbol{M}_2$ to denote the corresponding deterministic masks for $\boldsymbol{W}_1$ and $\boldsymbol{W}_2$, respectively. And we use $\boldsymbol{Z}_1$, $\boldsymbol{Z}_2$ to denote the corresponding random masks for $\boldsymbol{W}_1$ and $\boldsymbol{W}_2$, respectively. These linearly transform the layers' inputs before applying some element-wise non-linearity $\sigma()$. Denote by $\boldsymbol{b}$ the biases by which we shift the input of the non-linearity. We assume the model to output $D$ dimensional vectors while its input is $Q$ dimensional vectors, with $K$ hidden units. Thus $\boldsymbol{W}_1$, $\boldsymbol{M}_1$, and $\boldsymbol{Z}_1$ are $Q \times K$ matrices, $\boldsymbol{W}_2$, $\boldsymbol{M}_1$, and $\boldsymbol{Z}_1$ are $K \times D$ matrices, and $\boldsymbol{b}$ is a $K$ dimensional vector. A sparse NN model with the two masks would output $\widehat{y} = \sigma(\boldsymbol{x}\boldsymbol{Z}_1 \odot \boldsymbol{W}_1 + \boldsymbol{b})\boldsymbol{Z}_2 \odot \boldsymbol{W}_2$ given some input $\boldsymbol{x}$.

For the update of masks, the deterministic masks is updated by exploiting the magnitude of weights and gradients. The random mask is randomly sampled. For the update of weights, we use $E$ to denote the loss function, which is the euclidean loss for regression problem

$$E = \frac{1}{2N} \sum_{n=1} N||y_n - \widehat{y}_n||_2^2, \tag{6}$$

where $y_n$ is the observed response, $\widehat{y}_n$ is the prediction based on input $\boldsymbol{x}_n$ for $n = 1, \cdots, N$.

For classification task with D classes, we use softmax function to map the output $\widehat{y}_n$ to the probability score for each class $\widehat{p}_{nd} = \exp(\widehat{y}_{nd})/(\sum_k \exp(\widehat{y}_{nk}))$, and the loss function will be

$$E = -\frac{1}{N} \sum_{n=1} N \log(\widehat{p}_{n,c_n}), \tag{7}$$

where $c_n \in [1, 2, \cdots, D]$ is the true class label for $\boldsymbol{x}_n$.

During NN optimization process, apart from the loss function mentioned above, $l_2$ regularisation is often used to improve the performance, leading to a minimisation objective:

$$\mathcal{L}_{\text{CigL}} := \frac{1}{N} \sum_{i=1}^{N} E(y_i, \widehat{y}_i) + \lambda_1||\boldsymbol{W}_1||_2^2 + \lambda_2||\boldsymbol{W}_2||_2^2 + \lambda_3||\boldsymbol{b}||_2^2. \tag{8}$$

Therefore, duing the training process of CigL, we iteratively update the sparse weights $\boldsymbol{W}$ by minimising Eq. (8) and update deterministic mask based on the magnitude of weights and gradients.

## A.2  GAUSSIAN PROCESS

The Gaussian process (GP) is a popular non-parametric Bayesian methodto model distributions over functions, which can be applied to bothe regression and classification tasks. It has good performance in various fields, but it will bring huge computation burden when faced with a large number of data. The use of variational inference for GP can make it scalable to large data.

Given a data $\{\boldsymbol{x}_n, \boldsymbol{y}_n\}, n = 1, \cdots, N$, the task is to estimate an unknown function $\boldsymbol{y} = f(\boldsymbol{x})$, where $\boldsymbol{X} \in \mathbb{R}^{N \times Q}$ and $\boldsymbol{Y} \in \mathbb{R}^{N \times D}$. GP usually put a prior over the function space and we want to fit the posterior distribution over the function space:

$$p(\boldsymbol{f}|\boldsymbol{X}, \boldsymbol{Y}) \propto p(\boldsymbol{Y}|\boldsymbol{X}, \boldsymbol{f})p(\boldsymbol{f}).$$

Within Gaussian process, we usually place a Gaussian prior over the function space, and it equivalent to placing a joint Gaussian distribution over all function values

$$\boldsymbol{F}|\boldsymbol{X} \sim \mathcal{N}(\boldsymbol{0}, \boldsymbol{K}(\boldsymbol{X}, \boldsymbol{X})) \tag{9}$$

$$\boldsymbol{Y}|\boldsymbol{F} \sim \mathcal{N}(\boldsymbol{F}, \tau^{-1})$$

where $\tau$ is a precision parameter and $\boldsymbol{I}_N$ is the identity matrix with dimensions $N \times N$.

For classification tasks, we can formulate the model as

$$\boldsymbol{F}|\boldsymbol{X} \sim \mathcal{N}(\boldsymbol{0}, \boldsymbol{K}(\boldsymbol{X}, \boldsymbol{X})) \tag{10}$$

$$\boldsymbol{Y}|\boldsymbol{F} \sim \mathcal{N}(\boldsymbol{F}, \tau^{-1}),$$

$$c_n|\boldsymbol{Y} \sim \text{Categorical}\left( \exp(\widehat{y}_{nd})/(\sum_k \exp(\widehat{y}_{nk})) \right) \tag{11}$$

An important aspect of Gaussian process is the choice of the covariance function, which reflects how we believe the similarity between each pair of inputs $\boldsymbol{x}_i$ and $\boldsymbol{x}_j$. One widely-used covariance function is stationary squared exponential covariance function. In addition, some non-stationary covariance function are proposed, such as dot-product kernels and more flexible deep network kernels.

### A.3 HIERARCHICAL VARIATIONAL INFERENCE

Variational inference (VI) is a broadly-used technique to approximate intractable integrals in Bayesian modeling, which sets up a parameterized family of tractable distributions over the latent variables and then optimizes the parameters to be close to the posterior. More specifically, suppose $\boldsymbol{w}$ is the set of random variables defining our model. Then, the predictive distribution will be formulated as

$$p(\boldsymbol{y}^*|\boldsymbol{x}^*, \boldsymbol{X}, \boldsymbol{Y}) = \int p(\boldsymbol{y}^*|\boldsymbol{x}^*, \boldsymbol{w})p(\boldsymbol{w}|\boldsymbol{X}, \boldsymbol{Y})d\boldsymbol{w},$$

where the posterior $p(\boldsymbol{w}|\boldsymbol{X}, \boldsymbol{Y})$ is usually intractable. Thus, we define a family of tractable approximating variational distributions $q(\boldsymbol{w})$ to approach the posterior.

To find the closest approximating distribution among the family of $q(\boldsymbol{w})$, we minimise the Kullback–Leibler (KL) divergence between $q(\boldsymbol{w})$ and posterior $p(\boldsymbol{w}|\boldsymbol{X}, \boldsymbol{Y})$, which is equivalent to maximising the log evidence lower bound (ELBO) with respect to $q(\boldsymbol{w})$:

$$\mathcal{L}_{\text{VI}} := \int q(\boldsymbol{w}) \log p(\boldsymbol{Y}|\boldsymbol{X}, \boldsymbol{w})d\boldsymbol{w} - D_{\text{KL}}(q(\boldsymbol{w})\|p(\boldsymbol{w})).$$

After obtaining an good approximation $q(\boldsymbol{w})$, we can update the predictive distribution to

$$p(\boldsymbol{y}^*|\boldsymbol{x}^*, \boldsymbol{X}, \boldsymbol{Y}) = \int p(\boldsymbol{y}^*|\boldsymbol{x}^*, \boldsymbol{w})q(\boldsymbol{w})d\boldsymbol{w}.$$

However, when faced with posterior difficult to fit, $q(\boldsymbol{w})$ can be limited and not flexible enough to approach the posterior. In this case, VI cannot capture the posterior dependencies between latent variables that both improve the fidelity of the approximation and are sometimes intrinsically meaningful. To solve this limitation of VI, hierarchical variational inference (HVI) is proposed, which can capture both posterior dependencies between the latent variables and more complex marginal distributions (Ranganath et al., 2016). More specifically about HVI, we extend the limited family of VI distribution hierarchically, i.e., by placing a prior on the parameters of the likelihood. Suppose VI uses $q(\boldsymbol{w}; \lambda)$ to approximate the posterior where $\lambda$ is the variational parameters to optimise. HVI will add prior on $\lambda$ and uses $q(\boldsymbol{w}|\lambda)q(\lambda; \theta)$. The ELBO equivalently will be as:

$$\mathcal{L}_{\text{HVI}} := \mathbb{E}_{q_{\text{HVI}}(\boldsymbol{w}; \lambda)}[\log p(\boldsymbol{x}, \boldsymbol{w}) - \log q_{\text{HVI}}(\boldsymbol{w}; \theta)].$$

This ELBO can be further bounded by

$$\mathcal{L}_{\text{HVI}} \leq \mathbb{E}_{q_{\text{HVI}}(\boldsymbol{w}; \lambda)}[\log p(\boldsymbol{x}, \boldsymbol{w})] - \mathbb{E}_{q(\boldsymbol{w}, \lambda)}[\log q(\lambda) + \log q(\boldsymbol{w}|\lambda) - \log r(\lambda|\boldsymbol{w}; \theta)].$$

where $r(\lambda|\boldsymbol{w}; \theta)$ is introduced to apply the variational principle.

## B APPENDIX: CIGL AS A HIERARCHICAL BAYESIAN APPROXIMATION

We show that sparse deep NNs trained with CigL are mathematically equivalent to approximate hierarchical variational inference in the deep Gaussian process (marginalised over its covariance

function parameters). For this, we build on previous work (Gal & Ghahramani, 2016) that proved unit dropout applied before every weight layer are mathematically equivalent to approximate variational inference in the deep Gaussian process. Starting with the full Gaussian process we will develop an approximation that will be shown to be equivalent to the sparse NN optimisation objective with CigL (eq. (8)) with either the Euclidean loss in the case of regression or softmax loss in the case of classification. Our derivation takes regression as an example, which can be extended to classification by Section 4 of the Appendix of Gal & Ghahramani (2016). This view of CigL will allow us to derive new probabilistic results in sparse training.

### B.1    A GAUSSIAN PROCESS APPROXIMATION

In this section, we will re-parameterise the deep GP model and marginalise over the additional auxiliary random variables, which is built on Gal & Ghahramani (2016). To define our covariance function, let $\sigma(.)$ be some non-linear activation function and $\boldsymbol{K}(\boldsymbol{x}, \boldsymbol{y})$ can be formulated as:

$$\boldsymbol{K}(\boldsymbol{x}, \boldsymbol{y}) = \int p(\mathrm{w})p(\mathrm{b})\sigma(\mathrm{w}^\top \boldsymbol{x} + \boldsymbol{b})\sigma(\mathrm{w}^\top \boldsymbol{y} + \boldsymbol{b})d\boldsymbol{w}d\boldsymbol{b},$$

where $p(\mathrm{w})$ is a standard multivariate normal distribution in dimension $Q$.

We use Monte Carlo integration with K samples to approximate the integral above and get the finite rank covarinace function

$$\widehat{\boldsymbol{K}}(\boldsymbol{x}, \boldsymbol{y}) = \frac{1}{K} \sum_{k=1}^{K} \sigma(\mathrm{w}_k^\top \boldsymbol{x} + \boldsymbol{b}_k)\sigma(\mathrm{w}_k^\top \boldsymbol{y} + \boldsymbol{b}_k),$$

where $\mathrm{w}_k \sim p(\mathrm{w})$ and $\boldsymbol{b}_k \sim p(\mathrm{b})$. $K$ is the number of hidden units in our single hidden layer sparse NN approximation. The generative model will be as follow when we use $\widehat{\boldsymbol{K}}$ instead of $\boldsymbol{K}$:

$$\boldsymbol{w} \sim p(\boldsymbol{w}), \quad \mathrm{b}_k \sim p(\mathrm{b}),$$
$$\boldsymbol{W}_1 = [\boldsymbol{w}_{qk}]_{q=1}^{Q}{}_{k=1}^{K}, \quad \boldsymbol{b} = [b_k]_{k=1}^{K},$$
$$\widehat{\boldsymbol{K}}(\boldsymbol{x}, \boldsymbol{y}) = \frac{1}{K} \sum_{k=1}^{K} \sigma(\mathrm{w}_k^\top \boldsymbol{x} + \boldsymbol{b}_k)\sigma(\mathrm{w}_k^\top \boldsymbol{y} + \boldsymbol{b}_k),$$
$$\boldsymbol{F}|\boldsymbol{X}, \boldsymbol{W}_1, \boldsymbol{b} \sim \mathcal{N}(0, \widehat{\boldsymbol{K}}(\boldsymbol{x}, \boldsymbol{y})),$$
$$\boldsymbol{Y}|\boldsymbol{F} \sim \mathcal{N}(\boldsymbol{F}, \tau^{-1}\boldsymbol{I}_n),$$

where $\boldsymbol{W}_1 \in \mathbb{R}^{Q \times K}$ which parameterise the covariance function.

We can get the predictive distribution by integrating over the $\boldsymbol{F}$, $\boldsymbol{W}_1$, and $\boldsymbol{b}$

$$p(\boldsymbol{Y}|\boldsymbol{X}) = \int p(\boldsymbol{Y}|\boldsymbol{F})p(\boldsymbol{F}|\boldsymbol{W}_1, \boldsymbol{b}, \boldsymbol{X})p(\boldsymbol{W}_1)p(\boldsymbol{b})d\boldsymbol{W}_1 d\boldsymbol{b}.$$

Denoting a $1 \times K$ row vector

$$\phi(\boldsymbol{x}, \boldsymbol{W}_1, \boldsymbol{b}) = \sqrt{\frac{1}{K}}\sigma(\boldsymbol{W}_1^\top \boldsymbol{x} + \boldsymbol{b})$$

and a $N \times K$ feature matrix $\Phi = [\phi(\boldsymbol{x}_n, \boldsymbol{W}_1, \boldsymbol{b})]_{n=1}^{N}$. Then, we can get $\widehat{\boldsymbol{K}}(\boldsymbol{X}, \boldsymbol{X}) = \Phi\Phi^\top$ and the predictive distribution can be rewritten as

$$p(\boldsymbol{Y}|\boldsymbol{X}) = \int \mathcal{N}(\boldsymbol{Y}; 0, \Phi\Phi^\top + \tau^{-1}\boldsymbol{I}_N)p(\boldsymbol{W}_1)p(\boldsymbol{b})d\boldsymbol{W}_1 d\boldsymbol{b}$$

The normal distribution of $\boldsymbol{Y}$ inside the integral above can be written as a joint normal distribution over $\boldsymbol{y}_d$ which denoting the d-th columns of the $N \times D$ matrix $\boldsymbol{Y}$ $(d = 1, \cdots, D)$. For each term in the joint distribution, following Bishop & Nasrabadi (2006), we introduce a K × 1 auxiliary random variable $\boldsymbol{w}_d \sim \mathcal{N}(0, \boldsymbol{I}_K)$,

$$\mathcal{N}(\boldsymbol{y}_d; 0, \Phi\Phi^\top + \tau^{-1}\boldsymbol{I}_N) = \int \mathcal{N}(\boldsymbol{y}_d; \Phi\boldsymbol{w}_d, \tau^{-1}\boldsymbol{I}_N)\mathcal{N}(\boldsymbol{w}_d; 0, \boldsymbol{I}_K)d\boldsymbol{w}_d.$$

We use $\boldsymbol{W}_2 = [\boldsymbol{w}_d]_{d=1}^D \in \mathbb{R}^{K \times D}$ and we get the predictive distribution as

$$p(\boldsymbol{Y}|\boldsymbol{X}) = \int p(\boldsymbol{Y}|\boldsymbol{X}, \boldsymbol{W}_1, \boldsymbol{W}_2, \boldsymbol{b})p(\boldsymbol{W}_1)p(\boldsymbol{W}_2)p(\boldsymbol{b})d\boldsymbol{W}_1 d\boldsymbol{W}_2 d\boldsymbol{b}.$$

## B.2 HIERARCHICAL VARIATIONAL INFERENCE IN THE APPROXIMATE MODEL

We next approximate the posterior over these variables with appropriate hierarchical approximating variational distributions. We define a hierarchical variational distribution as:

$$q(\boldsymbol{W}_1, \boldsymbol{W}_2, \boldsymbol{b}) \coloneqq q(\boldsymbol{W}_1)q(\boldsymbol{W}_2)q(\boldsymbol{b}) = \int q(\boldsymbol{W}_1|\boldsymbol{U}_1)q(\boldsymbol{W}_2|\boldsymbol{U}_2)q(\boldsymbol{U}_1)q(\boldsymbol{U}_2)q(\boldsymbol{b})d\boldsymbol{U}_1 d\boldsymbol{U}_2,$$

where we define $q(\boldsymbol{W}_1)$ to be a Gaussian mixture distribution with two components, which is factorised over $Q$ and $K$:

$$q(\boldsymbol{W}_1|\boldsymbol{U}_1) = \prod_{q=1}^Q \prod_{k=1}^K q(\boldsymbol{w}_{qk}|\boldsymbol{u}_{qk}),$$

$$q(\boldsymbol{w}_{qk}|\boldsymbol{u}_{qk}) = p_1 \mathcal{N}(\boldsymbol{u}_{qk}, \sigma^2) + (1 - p_1)\mathcal{N}(0, \sigma^2),$$

$$q(\boldsymbol{u}_{qk}) = \mathcal{N}(\boldsymbol{v}_{qk}, \sigma^2)$$

where $p_1 \in [0, 1]$, and $\sigma > 0$. Similarly, we can define a hierarchical variational distribution over $\boldsymbol{W}_2$

$$q(\boldsymbol{W}_2|\boldsymbol{U}_2) = \prod_{k=1}^K \prod_{d=1}^D q(\boldsymbol{w}_{kd}|\boldsymbol{u}_{kd}),$$

$$q(\boldsymbol{w}_{kd}|\boldsymbol{u}_{kd}) = p_2 \mathcal{N}(\boldsymbol{u}_{kd}, \sigma^2) + (1 - p_2)\mathcal{N}(0, \sigma^2),$$

$$q(\boldsymbol{u}_{kd}) = \mathcal{N}(\boldsymbol{v}_{kd}, \sigma^2)$$

For $\boldsymbol{b}$, we use a simple Gaussian distribution

$$q(\boldsymbol{b}) = \mathcal{N}(\boldsymbol{u}, \sigma^2 \boldsymbol{I}_K).$$

## B.3 EVALUATING THE LOG EVIDENCE LOWER BOUND FOR REGRESSION

Next we evaluate the log evidence lower bound for the task of regression. The log evidence lower bound is as below

$$\mathcal{L}_{\text{GP-VI}} \coloneqq \int q(\boldsymbol{W}_1, \boldsymbol{W}_2, \boldsymbol{b}) \log p(\boldsymbol{Y}|\boldsymbol{X}, \boldsymbol{W}_1, \boldsymbol{W}_2, \boldsymbol{b}) - D_{\text{KL}}(q(\boldsymbol{W}_1, \boldsymbol{W}_2, \boldsymbol{b})\|p(\boldsymbol{W}_1, \boldsymbol{W}_2, \boldsymbol{b})).$$

where the integration is with respect to $\boldsymbol{W}_1, \boldsymbol{W}_2, \boldsymbol{b}$.

During regression, we can rewrite the integrand as a sum:

$$\log p(\boldsymbol{Y}|\boldsymbol{W}_1, \boldsymbol{W}_2, \boldsymbol{b}) = \sum_{d=1}^D \log \mathcal{N}(\boldsymbol{y}_d; \Phi \boldsymbol{w}_d, \tau^{-1}\boldsymbol{I}_N),$$

$$= -\frac{ND}{2}\log(2\pi) + \frac{ND}{2}\log(\tau) - \sum_{d=1}^D \frac{\tau}{2}\|\boldsymbol{y}_d - \Phi \boldsymbol{w}_d\|_2^2.$$

as the output dimensions of a multi-output Gaussian process are assumed to be independent. Denote $\widehat{\boldsymbol{Y}} = \Phi \boldsymbol{W}_2$. We can then sum over the rows instead of the columns of $\widehat{\boldsymbol{Y}}$ and write

$$\sum_{d=1}^D \frac{\tau}{2}\|\boldsymbol{y}_d - \widehat{\boldsymbol{y}}_d\|_2^2 = \sum_{n=1}^N \frac{\tau}{2}\|\boldsymbol{y}_n - \widehat{\boldsymbol{y}}_n\|_2^2.$$

Here we have $\widehat{\boldsymbol{y}}_n = \phi(\boldsymbol{x}, \boldsymbol{W}_1, \boldsymbol{b})\boldsymbol{W}_2 = \sqrt{\frac{1}{K}}\sigma(\boldsymbol{x}_n\boldsymbol{W}_1 + \boldsymbol{b})\boldsymbol{W}_2$, leading to

$$\log p(\boldsymbol{Y}|\boldsymbol{W}_1, \boldsymbol{W}_2, \boldsymbol{b}) = \sum_{n=1}^N \log \mathcal{N}(\boldsymbol{y}_n; \phi(\boldsymbol{x}_n, \boldsymbol{W}_1, \boldsymbol{b})\boldsymbol{W}_2, \tau^{-1}\boldsymbol{I}_D).$$

Therefore, we can update the log evidence lower bound as

$$\sum_{n=1}^{N} \int q(\boldsymbol{W}_1, \boldsymbol{W}_2, \boldsymbol{b}) \log p(\boldsymbol{Y}|\boldsymbol{x}_n, \boldsymbol{W}_1, \boldsymbol{W}_2, \boldsymbol{b}) - D_{\text{KL}}(q(\boldsymbol{W}_1, \boldsymbol{W}_2, \boldsymbol{b})\|p(\boldsymbol{W}_1, \boldsymbol{W}_2, \boldsymbol{b})).$$

We re-parametrise the integrands in the sum to not depend on $\boldsymbol{W}_1, \boldsymbol{W}_2, \boldsymbol{b}$ directly, but instead on the standard normal distribution and the Bernoulli distribution. Let $q(\epsilon_1) = \mathcal{N}(0, \boldsymbol{I}_{Q \times K})$, $q(\mathbf{z}_{1,q,k}) = \text{Bernoulli}(p_1)$, $q(\epsilon_2) = \mathcal{N}(0, \boldsymbol{I}_{K \times D})$, $q(\mathbf{z}_{2,k,d}) = \text{Bernoulli}(p_2)$, $q(\epsilon) = \mathcal{N}(0, \boldsymbol{I}_K)$, $q(\epsilon_3) = \mathcal{N}(0, \boldsymbol{I}_{Q \times K})$, and $q(\epsilon_4) = \mathcal{N}(0, \boldsymbol{I}_{K \times D})$. Then, we can have

$$\boldsymbol{W}_1 = \boldsymbol{Z}_1 \odot (\boldsymbol{U}_1 + \sigma \epsilon_1) + (1 - \boldsymbol{Z}_1) \odot \sigma \epsilon_1,$$
$$\boldsymbol{W}_2 = \boldsymbol{Z}_2 \odot (\boldsymbol{U}_2 + \sigma \epsilon_2) + (1 - \boldsymbol{Z}_2) \odot \sigma \epsilon_2,$$
$$\boldsymbol{b} = \boldsymbol{u} + \sigma \epsilon, \ \boldsymbol{U}_1 = \sigma \epsilon_3, \ \boldsymbol{U}_2 = \sigma \epsilon_4$$

where $\odot$ means element-wise multiplication. Thus, we can update the above the sum over the integrals

$$\sum_{n=1}^{N} \int q(\boldsymbol{W}_1, \boldsymbol{W}_2, \boldsymbol{b}) \log p(\boldsymbol{y}_d|\boldsymbol{x}_n, \boldsymbol{W}_1, \boldsymbol{W}_2, \boldsymbol{b}),$$

$$= \sum_{n=1}^{N} \int q(\boldsymbol{Z}_1, \epsilon_1, \boldsymbol{Z}_2, \epsilon_2, \epsilon, \epsilon_3, \epsilon_4) \log p(\boldsymbol{y}_d|\boldsymbol{x}_n, \boldsymbol{W}_1(\boldsymbol{Z}_1, \epsilon_1, \epsilon_3), \boldsymbol{W}_2(\boldsymbol{Z}_2, \epsilon_2, \epsilon_4), \boldsymbol{b}(\epsilon)).$$

For the first term in $\mathcal{L}_{\text{GP-MC}}$, we can estimate the integrals using Monte Carlo integration with a distinct single sample to obtain

$$\mathcal{L}_{\text{GP-MC}} := \sum_{n=1}^{N} \int \log p(\boldsymbol{y}_d|\boldsymbol{x}_n, \widehat{\boldsymbol{W}}_1^n, \widehat{\boldsymbol{W}}_2^n, \widehat{\boldsymbol{b}}^n) - D_{\text{KL}}(q(\boldsymbol{W}_1, \boldsymbol{W}_2, \boldsymbol{b})\|p(\boldsymbol{W}_1, \boldsymbol{W}_2, \boldsymbol{b})).$$

Following Gal & Ghahramani (2016), by optimising the stochastic objective $\mathcal{L}_{\text{GP-MC}}$, we can converge to the same limit as $\mathcal{L}_{\text{GP-VI}}$, which justifies this stochastic approximation.

Moving to the second term in $\mathcal{L}_{\text{GP-MC}}$, we use w and u to denote certain component in the weights ($\boldsymbol{W}_1$) and variational parameters ($\boldsymbol{U}_1$), respectively. Then, we can have

$$-D_{\text{KL}}\left(q(\mathbf{w})\|p(\mathbf{w})\right) = -\int q(\mathbf{w}) \log \frac{q(\mathbf{w})}{p(\mathbf{w})} d\mathbf{w}$$

$$= \int \int q(\mathbf{w}, \mathbf{u}) d\boldsymbol{u} \log \frac{p(\mathbf{w})}{\int q(\mathbf{w}, \mathbf{u}) d\mathbf{u}} d\mathbf{w}$$

$$= \int q(\mathbf{u}) \{ \int q(\mathbf{w}|\mathbf{u}) \log \frac{p(\mathbf{w})}{\int q(\mathbf{w}, \mathbf{u}) d\mathbf{u}} d\mathbf{w} \} d\mathbf{u}$$

$$= \int q(\mathbf{u}) [ \int q(\mathbf{w}|\mathbf{u}) \log p(\mathbf{w}) d\mathbf{w} ] d\mathbf{u} - \int q(\mathbf{u}) \{ \int q(\mathbf{w}|\mathbf{u}) \log [\int q(\mathbf{w}, \mathbf{u}) d\mathbf{u}] d\mathbf{w} \} d\mathbf{u}$$

$$= \int q(\mathbf{u}) [ \int q(\mathbf{w}|\mathbf{u}) \log p(\mathbf{w}) d\mathbf{w} ] d\mathbf{u} - \int [ \int q(\mathbf{w}, \mathbf{u}) d\mathbf{u}] \log [\int q(\mathbf{w}, \mathbf{u}) d\mathbf{u}] d\mathbf{w} \qquad (12)$$

For the first term in Eq. (12), we can first follow the Proposition 1 in (Gal & Ghahramani, 2016) to approximate $\int q(\mathbf{w}|\mathbf{u}) \log p(\mathbf{w}) d\mathbf{w}$ as below:

$$\int q(\mathbf{w}|\mathbf{u}) \log p(\mathbf{w}) d\mathbf{w} \approx -\frac{1}{2}(p \cdot \mathbf{u}^2 + \sigma^2)$$

Then, we can approximate the first term in Eq. (12) as

$$\int q(\mathbf{u}) [ \int q(\mathbf{w}|\mathbf{u}) \log p(\mathbf{w}) d\mathbf{w} ] d\mathbf{u} \approx \int q(\mathbf{u}) [-\frac{1}{2}(p \cdot \mathbf{u}^2 + \sigma^2)] d\mathbf{u}$$

$$= -\frac{1}{2}\sigma^2 - \frac{p}{2} \int q(\mathbf{u}) \mathbf{u}^2 d\mathbf{u} = -\frac{p+1}{2}\sigma^2 - \frac{p}{2}\mathbf{v}^2$$

For the second term in Eq. (12), we can estimate the integral $\int q(\mathrm{w}, \mathrm{u})d\mathrm{u}$ using Monte Carlo integration with a distinct single sample to obtain

$$\int [\int q(\mathrm{w}, \mathrm{u})d\mathrm{u}] \log[\int q(\mathrm{w}, \mathrm{u})d\mathrm{u}]d\mathrm{w} \approx \int q(\mathrm{w}|\widehat{\mathrm{u}}) \log[q(\mathrm{w}|\widehat{\mathrm{u}})]d\mathrm{w}$$

Then we can follow the Proposition 1 in (Gal & Ghahramani, 2016) to approximate it as

$$\int q(\mathrm{w}|\widehat{\mathrm{u}}) \log[q(\mathrm{w}|\widehat{\mathrm{u}})]d\mathrm{w} \approx \frac{1}{2}(\log \sigma^2 + 1 + 2\log \pi) + C$$

Therefore, we can get the approximation for Eq. (12) as Eq. (13):

$$-D_{\mathrm{KL}}\left(q(\mathrm{w})\|p(\mathrm{w})\right) = -\frac{p+1}{2}\sigma^2 - \frac{p}{2}\mathrm{v}^2 + \frac{1}{2}(\log \sigma^2 + K(1 + 2\log \pi)) + C \qquad (13)$$

Then, we can have the following equation based on Eq. (13):

$$D_{\mathrm{KL}}(q(\boldsymbol{W}_1)\|p(\boldsymbol{W}_1)) \approx \frac{QK(p+1)}{2}\sigma^2 - \frac{QK}{2}(\log(\sigma^2) + 1) + \frac{p_1}{2}\sum_{q=1}^{Q}\sum_{k=1}^{K}\mathrm{v}_{qk}^2 + C,$$

where $C$ is a constant and $D_{\mathrm{KL}}(q(\boldsymbol{W}_2)\|p(\boldsymbol{W}_2))$ can be approximated in a similar way. For $D_{\mathrm{KL}}(q(\boldsymbol{b})\|p(\boldsymbol{b}))$, it can be written as

$$D_{\mathrm{KL}}(q(\boldsymbol{b})\|p(\boldsymbol{b})) = \frac{1}{2}(\boldsymbol{u}^\top\boldsymbol{u} + K(\sigma^2 - \log(\sigma^2) - 1)) + C.$$

### B.4 Log Evidence Lower Bound Optimisation for CigL

Next we explain the relation between the above equations with equations for CigL. Ignoring the constant terms $\tau, \sigma$ we obtain the maximisation objective

$$\mathcal{L}_{\text{GP-MC}} \propto -\frac{\tau}{2}\sum_{n=1}^{N}||\boldsymbol{y}_n - \widehat{\boldsymbol{y}}_n||_2^2 - \frac{p_1}{2}||\boldsymbol{V}_1||_2^2 - \frac{p_2}{2}||\boldsymbol{V}_2||_2^2 - \frac{1}{2}||\boldsymbol{u}||_2^2. \qquad (14)$$

We will show the equivalence between the iterative update of $\boldsymbol{M}_1$, $\boldsymbol{Z}_1$ and $\boldsymbol{U}_1$ in CigL and the hierarchical variational inference for deep GP. The update for $\boldsymbol{M}_2$, $\boldsymbol{Z}_2$ and $\boldsymbol{U}_2$ will be similar. In the hierarchical variational distribution, the distribution of sparse weights $\boldsymbol{W}_1$ depends on three random variables $\boldsymbol{M}_1$, $\boldsymbol{Z}_1$, and $\boldsymbol{U}_1$. Given $\boldsymbol{Z}_1$ and $\boldsymbol{U}_1$, we can know the variational distribution for $\boldsymbol{M}_1$ as:

$$q(\boldsymbol{M}_1|\boldsymbol{U}_1) \propto \exp(\boldsymbol{M}_1 \odot |\boldsymbol{U}_1|) \qquad (15)$$

where $\boldsymbol{M}_1$ is under certain sparsity constraint. Thus, we can update $\boldsymbol{M}_1$ by choosing $\widehat{\boldsymbol{M}}_1$ that maximising Eq. (15), which is aligned with the $\boldsymbol{M}_1$ update procedure in CigL.

Given $\boldsymbol{M}_1$, we can use $\widehat{\boldsymbol{V}}_1$ to approximate $\widehat{\boldsymbol{U}}_1$. Then, we can let $\sigma$ tend to zero (Gal & Ghahramani, 2016), and the random variable realisations $\widehat{\boldsymbol{W}}_1^n, \widehat{\boldsymbol{W}}_2^n, \widehat{\boldsymbol{b}}^n$ can be

$$\widehat{\boldsymbol{W}}_1^n \approx \widehat{\boldsymbol{Z}}_1 \odot \widehat{\boldsymbol{U}}_1, \quad \widehat{\boldsymbol{W}}_2^n \approx \widehat{\boldsymbol{Z}}_2 \odot \widehat{\boldsymbol{U}}_2, \quad \widehat{\boldsymbol{b}}^n \approx \widehat{\boldsymbol{u}}$$

Then, we can get

$$\widehat{\boldsymbol{y}}_n \approx \sqrt{\frac{1}{K}}\sigma(\boldsymbol{x}_n(\widehat{\boldsymbol{Z}}_1 \odot \widehat{\boldsymbol{U}}_1) + \widehat{\boldsymbol{u}})(\widehat{\boldsymbol{Z}}_2 \odot \widehat{\boldsymbol{U}}_2).$$

We scale the optimisation objective by a positive constant $\frac{1}{\tau N}$ and get the objective:

$$\mathcal{L}_{\text{GP-MC}} \propto -\frac{1}{2N}\sum_{n=1}^{N}||\boldsymbol{y}_n - \widehat{\boldsymbol{y}}_n||_2^2 - \frac{p_1}{2\tau N}||\boldsymbol{U}_1||_2^2 - \frac{p_2}{2\tau N}||\boldsymbol{U}_2||_2^2 - \frac{1}{2\tau N}||\boldsymbol{u}||_2^2.$$

So, we recover equation for CigL. With correct stochastic optimisation scheduling, both will converge to the same limit.

### B.5 Mask & Weight Averaging for Prediction

For prediction, we design mask & weight averaging (WMA) to produce the final output model. Specifically, we collect samples of $\{\widehat{\boldsymbol{Z}}_1, \widehat{\boldsymbol{U}}_1\}$ during the optimization process, where we use $\widehat{\boldsymbol{V}}_1$ at different epochs to approximate $\widehat{\boldsymbol{U}}_1$. By using weight & mask averging and letting $\sigma$ tend to zero (Gal & Ghahramani, 2016), the random variable $\widehat{\boldsymbol{W}}_1, \widehat{\boldsymbol{W}}_2, \widehat{\boldsymbol{b}}$ can be approximated as

$$\widehat{\boldsymbol{W}}_1 \approx \frac{1}{S}\sum_{s=1}^{S} \widehat{\boldsymbol{Z}}_1^{(s)} \odot \widehat{\boldsymbol{U}}_1^{(s)}, \quad \widehat{\boldsymbol{W}}_2 \approx \frac{1}{S}\sum_{s=1}^{S} \widehat{\boldsymbol{Z}}_2^{(s)} \odot \widehat{\boldsymbol{U}}_2^{(s)}, \quad \widehat{\boldsymbol{b}} \approx \frac{1}{S}\sum_{s=1}^{S} \widehat{\boldsymbol{u}}^{(s)}.$$

WMA can be seen as approximating the mean of the posterior based on samples from variational distribution $q(\boldsymbol{W})$ using moment matching, which is justified as below:

(i) CigL is connected to the Bayesian approach because of using both deterministic and random masks to explore the weight space. As shown in Equation 3, the design of $\boldsymbol{Z}$ and $\boldsymbol{M}$ results in a hierarchical variational distribution $q(\boldsymbol{W})$, where the hierarchy expands the approximation family and leads to a better posterior approximation capability. In Equation 3, updating the mask is equivalent to updating and to bring closer to the posterior 3.

(ii) In a similar idea to weight dropout (Gal & Ghahramani, 2016), WMA can be considered as a Bayesian approximation, which is used to approximate the mean of the posterior by moment matching.

- Weight dropout has been shown to be equivalent to the Bayesian approximation method (Gal & Ghahramani, 2016). After obtaining the final $\boldsymbol{W}$, dropout approximates $\int f(\boldsymbol{W}\boldsymbol{Z})p(\boldsymbol{Z})d\boldsymbol{Z}$ using $f(E(\boldsymbol{W}\boldsymbol{Z}))$, where $f$ is the neural network and the mean $E(\boldsymbol{W}\boldsymbol{Z}) = \int \boldsymbol{W}\boldsymbol{Z}p(\boldsymbol{Z})d\boldsymbol{Z}$ (Srivastava et al., 2014). This approximation actually approximates the whole posterior by the first moment of the posterior (i.e., the mean).

- For our WMA, since we assume a hierarchy, we collect multiple samples of $\boldsymbol{W}$ and $\boldsymbol{Z}$. Then, the WMA is used to approximate the first moment of the posterior which is used as an approximation of the posterior itself (Srivastava et al., 2014).

- In addition, if we really want the second moment, it is straightforward to obtain an estimation based on samples using moment matching again, similar to Maddox et al. (2019). We do not estimate the second moment since the sparse training typically wants a single sparse model in the end to reduce both computational and memory costs, and we find that using the posterior mean already significantly improves the calibration of the sparse training.

## C Appendix: Additional Experimental Results

### C.1 Stronger Correlation Between Hidden Variables

Empirically, we find that a stronger correlation between $\boldsymbol{Z}$ and $\boldsymbol{W}$ in sparse training. We use CigL to train sparse Wide-ResNet-22-2 on CIFAR-10 at multiple sparsities (0%, 50%, 80%, 90%). Then, we randomly draw five random masks $\boldsymbol{Z}_i, i \in 1, \cdots, 5$ from Bernoulli distribution. Using the final sparse weights $\boldsymbol{W}$, we obtain several new sparse models $\boldsymbol{Z}_i \odot \boldsymbol{W}, i \in 1, \cdots, 5$, and record their test accuracies. We compare the test accuracy of $\boldsymbol{W}$ with the average accuracy of $\boldsymbol{Z}_i \odot \boldsymbol{W}, i \in 1, \cdots, 5$ to see the correlation. The larger decrease in test accuracy after multiplying by $\boldsymbol{Z}_i$ implies a stronger correlation between $\boldsymbol{Z}$ and $\boldsymbol{W}$.

As shown in Figure 6 (a), both the accuracy of $\boldsymbol{W}$ (red curve) and the average accuracy of $\boldsymbol{Z}_i \odot \boldsymbol{W}, i \in 1, \cdots, 5$ (blue curve) are decrease with increasing sparsity, and we see a more pronounced decrease in the blue curve. Figure 6 (b) further shows the decrease in test accuracy at each sparsity. We observe that the decrease is very small in the dense or sparse model at low sparsity. However, when it shifts to high sparsity such as 90%, we observe a larger decrease, which indicates a stronger correlation between $\boldsymbol{Z}$ and $\boldsymbol{W}$ in sparse training.

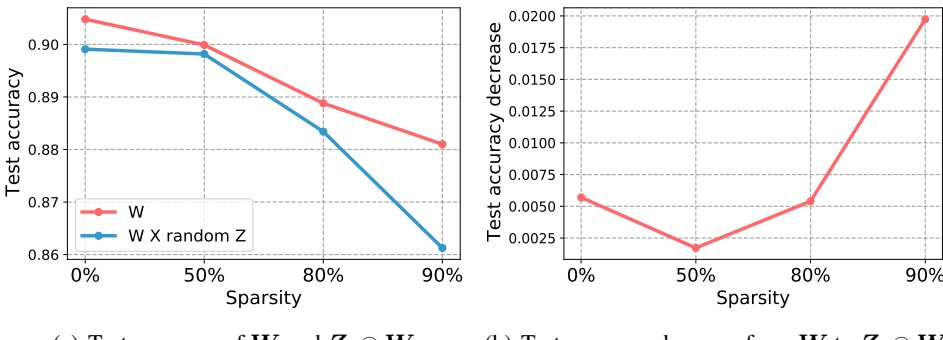

(a) Test accuracy of $\boldsymbol{W}$ and $\boldsymbol{Z}_i \odot \boldsymbol{W}$          (b) Test accuracy decrease from $\boldsymbol{W}$ to $\boldsymbol{Z}_i \odot \boldsymbol{W}$

Figure 6: (a) Test accuracy of sparse model $\boldsymbol{W}$ and the newly produced sparse model $\boldsymbol{Z}_i \odot \boldsymbol{W}$. (b) decrease in test accuracy from sparse model $\boldsymbol{W}$ to newly produced sparse model $\boldsymbol{Z}_i \odot \boldsymbol{W}$. At low sparsity, the decrease of the dense or sparse models is small. At high sparsity, the decrease is larger.

## C.2 RELIABILITY IN SPARSE TRAINING

To get a more comprehensive understanding of the reliability issues in sparse training, we also evaluated the ECE values of the sparse models generated by SET (Mocanu et al., 2018). We find that the sparse model produced by SET is also more over-confident than the dense model. As shown in Table 4, the ECE values of dense ResNet-50 are smaller than those of sparse ResNet-50 on both CIFAR-10 and CIFAR-100. This proves the reliability issue of sparse training.

Table 4: ECE value of sparse ResNet-50 on CIFAR-10 and CIFAR-100 produced by SET at different sparsity including 0%, 50%, 80%, 90%, 95%, 99%.

| SPARSITY | 0% | 50% | 80% | 90% | 95% | 99% |
|---|---|---|---|---|---|---|
| CIFAR-100 | 0.0381 | 0.0429 | 0.0416 | 0.0459 | 0.0460 | 0.0589 |
| CIFAR-100 | 0.0841 | 0.0931 | 0.1058 | 0.1290 | 0.1282 | 0.0873 |

## C.3 MORE COMPARISON WITH SPARSE TRAINING BASELINE

We further compare our CigL with a recent Sparse training baseline Sup-tickets (Yin et al., 2022) to show the effectiveness of CigL in reducing ECE values. Table 5 shows the change in ECE after using Sup-tickets or our CigL. We can see that Sup-tickets brings only a limited reduction in ECE, while the reduction of our CigL is much larger than that of Sup-tickets.

Table 5: ECE value changes of Sup-tickets and CigL in ResNet-50 on CIFAR-10 and CIFAR-100 at different sparsity including 80%, 90%, 95%.

| | CIFAR-10 | | | CIFAR-100 | | |
|---|---|---|---|---|---|---|
| | 80% | 90% | 95% | 80% | 90% | 95% |
| SUP-TICKETS | -0.0012 | -0.0005 | -0.0007 | -0.0005 | -0.0010 | -0.0010 |
| CIGL | **-0.0067** | **-0.0080** | **-0.0119** | **-0.0141** | **-0.0113** | **-0.0104** |

## C.4 MORE RESULTS ABOUT THE EFFECT OF WEIGHT & MASK AVERAGING

To demonstrate that weight & mask averaging (WMA) is not effective in reducing ECE alone, we add more results of using only WMA without random masking (CigL w/o RM). Table 6 shows the change in ECE after using CigL w/o RM or our CigL. We find that when only WMA is used,

the ECE value cannot be effectively reduced, either increasing or with limited reduction. On the contrary, the reduction of our CigL is much larger than that of CigL w/o RM.

Table 6: ECE value changes of CigL w/o RM and CigL in ResNet-50 (CIFAR-100) and Wide-ResNet-22-2 (CIFAR-10) at different sparsity including 80%, 90%, 95%, 99%.

| | RESNET-50, CIFAR-100 | | | | WIDE-RESNET-22-2, CIFAR-10 | | | |
|---|---|---|---|---|---|---|---|---|
| | 80% | 90% | 95% | 99% | 80% | 90% | 95% | 99% |
| CIGL W/O RM | 0.0038 | -0.0098 | -0.0144 | -0.0046 | 0.0060 | 0.0129 | 0.0149 | 0.0017 |
| CIGL | **-0.0010** | **-0.0152** | **-0.0344** | **-0.0269** | **-0.0141** | **-0.0113** | **-0.0104** | **-0.0049** |

## D  MORE DISCUSSION

### D.1  WEIGHT SPACE EXPLORATION

ITOP (Liu et al., 2021) and DST-EE (Huang et al., 2022) study weight space exploration in sparse training and emphasized its importance. Compared to their studies, our work has two main differences that address their limitations.

On the one hand, our work has a different goal from ITOP and DST-EE with respect to encouraging exploration of the weight space. Specifically, our work aims to better explore the weight space to find more reliable models, while ITOP and DST-EE aims to build models with higher accuracy, ignoring the safety aspects.

On the other hand, the exploration of weight space has two aspects, namely "which weight is active" & "what value that weight has". The limitation of ITOP is that, given the mask, the second aspect is not addressed and the optimization of the algorithm remains more challenging than dense training due to the pseudo-local optimization introduced by the sparsity constraints. To meet this challenge, ITOP increases the iterations between mask updates, leading to an increase in training time. For DST-EE, it mainly targets the first aspect. In contrast to their study, our work addresses this limitation, as shown in the following discussion:

The first aspect of weight space exploration is reflected by the ITOP rate, which is the percentage of all weights that have ever been selected as active weights by the mask. The second aspect of weight space exploration is reflected by the idea of "reliable exploration" in the ITOP paper. Ideally, a reliable exploration should allow a model to find the good direction and jump out of the bad local optimum. The sparsity constraints introduce some pseudo-local optima, which is difficult to jump out of. Our random mask can randomly cut off some directions and force the model to explore other directions, thus encouraging the model to better explore the weight space and avoid missing the correct direction.

### D.2  DOUBLE DESCENT IN RELIABILITY

One phenomenon we find worth discussing is the *double descent* in the reliability of sparse training. We discuss it in Section 6, where we divide the sparsity into four stages, i.e., poor model, shallow model, sparse deep model, and dense deep model.

The four stages are first supported by intuition. In the discussion, we draw analogies between model types such as "shallow models" and "poor models" in terms of model accuracy (expressiveness) and size. Consistent with the previous definition of double descent (Nakkiran et al., 2021; Somepalli et al., 2022), we consider sparsity as a measure of model size. Intuitively, as we gradually reduce the model size (increase the sparsity), we will go through four stages.

The four arguments are also supported by our sparse training experiments on ResNet-50 at CIFAR-100. As shown in Figure 1 (c), we can infer the model type by sparsity and accuracy:

• For 99.7% sparsity, the accuracy of the model is 41.7%, which is similar to a shallow model.

- For 99.9% sparsity, the accuracy of the model is 23.5%, which can be viewed as a poor model.

More detailed and quantitative support for these four arguments is beyond the main scope of this paper and could be a good direction for future research. One potential direction is the use of effective depth as a measure of stage identification.

### D.3  WEIGHT & MASK AVERAGING

Without the use of WMA, the analysis in Section 4.2 would be a non-hierarchical Bayesian method or a poor approximation to a hierarchical Bayesian approach.

(i) In the absence of WMA, the algorithm can be viewed as a non-hierarchical variational inference. As described in Section 4.3, using the final $W$ for prediction without WMA is equivalent to using weight dropout in RigL. Thus, the analysis in Section 4.2 will be updated in a similar way to Section 3 in Gal & Ghahramani (2016) which shows that weight dropout can be viewed as a non-hierarchical Bayesian approximation.

(ii) Without WMA, the algorithm can also be viewed as a poor approximation to hierarchical variational inference.

- If we continue to interpret the algorithm without WMA using the current analysis structure from Section 4.1, then how we generate the final posterior approximation will change.

- In this case, although the algorithm is still a Bayesian approximation, we only use the final $W$ to represent the posterior, which does not effectively capture all the information we explore from the weight space and the increased correlation between $Z$ and $W$.

- Therefore, it turns out to be a bad hierarchical approximation, which limits its power.

### D.4  MULTI-MASK SPARSE DNNs

Existing multi-mask methods are not designed for improved weight space exploration. Bibikar et al. (2022) considers sparse training in federated learning and investigates the aggregation of multiple masks in edge devices. Xia et al. (2022) utilizes multiple masks with different granularities to allow greater flexibility in structured pruning and to improve accuracy. Despite the use of multiple masks, existing work (Xia et al., 2022; Bibikar et al., 2022) differ significantly from our work. They still use deterministic masks, which still suffer from the lack of exploration of the weight space, and consider only the accuracy of sparse models. In addition, their setups are federated learning and pruning, which are different from our work.

### D.5  SPARSE DNNs: PRUNING & SPARSE TRAINING

Although pruning (e.g., Lottery Tickets) and sparse training are related and both produce subnetworks with high accuracy, their goals and discovering journeys are quite different, which leads to significant differences in several important properties, including uncertainty, geometry of the loss surface, generalization ability, and so on.

(i) For the goal, Lottery Tickets mainly aim to reduce the inference cost, while the sparse training also aims to save resources during the training phase.

(ii) Lottery Tickets and sparse training are different in several important properties. As shown in Figure 11 of Chen et al. (2022), the blue and purple bars represent Lottery Tickets and sparse training with a sparsity level of 79%, respectively.

- For uncertainty, sparse training does not improve the confidence calibration compared to dense training, while Lottery Tickets allows for improved confidence calibration.

- For the geometry of the loss surface, sparse training leads to larger trace values and cannot locate flat local minima. In contrast, Lottery Tickets can still locate flat local minima.

- For generalization ability, sparse training provides higher accuracy and improved robustness compared to dense training, while Lottery Tickets provide relatively less improvement.

(iii) The main reason for the different properties is their different discovering journeys.

For Lottery Tickets:

• It retrains the weights from the initial training phase after each pruning, which significantly increases the training time but allows more time for the model to explore the weight space.

• It starts from a dense model and has low sparsity in the early stages, which reduces the difficulty of weight space exploration caused by the sparsity constraints.

For spare training:

• It maintains a high level of sparsity throughout the training process, which does not extend the training time to enable more exploration of the weight space.

• In addition, maintaining high sparsity can cut off a large portion of the optimization route and produce more spurious local minima, thus making training very difficult.

• Chen et al. (2022) shows the differences in the properties of Lottery Tickets and sparse training at the 79% sparsity level. The difficulty of training typically increases with increasing sparsity, implying that the difference is likely to be greater at higher sparsity levels.

