# OpenReview forum: "Calibrating the Rigged Lottery: Making All Tickets Reliable"
_ICLR.cc/2023/Conference — ICLR 2023 poster_

### Official Review · Reviewer_db8U · 2022-10-22

**Confidence:** 4
**Correctness:** 2
**Technical Novelty And Significance:** 4
**Empirical Novelty And Significance:** 4
**Recommendation:** 8

**Clarity, Quality, Novelty And Reproducibility:**

Overall a well-written and well-motivated paper with a novel (as far as I'm aware) and somewhat intuitive method for addressing calibration with sparse training. Some notable issues on the clarify front however:

Major Issues:
* ECE acronym is not defined until page 6
* ADOPT seems to be used instead of CiGL in a number of the figures/tables, without being explained. I'm assuming that it's just an old name for CiGL that stuck around...
* There is much more focus on the existence of a proof that there is an equivalence between CiGL and variational approximations of a deep Gaussian process itself than motivating why this should matter to the reader! Section 4.2. should be the introduction to 4.1 to address this, and in the introduction a brief statement based on 4.2 should be added to motivate this contribution.
* Multiple times it's claimed that both the model weights and masks are "averaged", and this is true, but the way this is written it led myself at least to believe these operations were independent, and in particular wonder what "mask averaging" is/can be effected. In Algorithm 1 a (joint) running average of the masked weights is performed, and it appears that this is what is being discussed. Although it's possible I'm in the minority in how I read this, I would reword this to avoid giving the reader the impression these are independent.

Minor:
* I would mention, briefly, the connection with dropout earlier in the introduction if only to mention to the reader that it's discussed later in the paper.

**Strength And Weaknesses:**

## Strengths:
* Overall a well-written and well-motivated paper with a novel and intuitive method for addressing calibration with sparse training.
* As far as I'm aware, the authors are the first to identify and study the problem of calibration in sparse training and RiGL specifically (but see below on the "comprehensive" claim).
* Explicitly identifies and discusses the connection with dropout that alert readers will be questioning as soon as they start reading the paper.
* The authors present experiments comparing their method, CiGL, with a popular dynamic sparse training method CiGL is based on, RiGL, and show comparable accuracy across a range of datasets and models.
* The authors present experiments showing that the Expected Calibration Error (ECE) of the proposed CiGL method is consistently and significantly lower than both RiGL, and existing methods of calibrating neural networks applied to RiGL.
* The authors present ablation studies to explore the importance of different components of their method.
* The results are in general very well presented, in particular Figure 4 is a brilliant way of visualizing the relative calibration of each of the methods at different sparsities.
* The method is well-motivated by comparing it with variational approximation of deep Gaussian processes, and dropout, although the former could be done better (see below).

## Weaknesses:
* Massive over-claim on the experimental front that really weakens what is otherwise a strong set of contributions. Multiple times in the paper it's claimed this is the first "comprehensive study on the reliability of sparse training...", and general statements are made about sparse training methods in general having issues with calibration/over-confidence. However the paper only looks at one sparse training method: RiGL (not even SET) as far as I can see. While the results there are indeed enough to motivate the author's method, and to make the community question the over-confidence of other sparse training methods, they are far from comprehensive enough to claim this applies to *all sparse training methods*. As far as I'm aware, the authors are the first to identify and study the problem of calibration in sparse training, and this would be a much better supported (and very strong) claim to make already.
* In Algorithm 1, the specific pseudocode used for updating the joint weight/mask running mean, i.e. $W_{CigL}=\frac{W_{CigL}\cdot n_{models}+W^{(t)} \odot Z}{n_{models}+1}$ is known for being extremely numerically unstable method in practice to calculate a running mean, and will for any moderate number of time steps $t$ results in a significant loss in floating point precision. The authors should detail if they used a numerically stable algorithm (e.g. Welford's algorithm) for this in their code, and if not repeat experiments with such a method or explain why their method is not sensitive to the expected loss in precision.
* In section 2.1 the authors claim that "Existing sparse training methods use only one mask to determine the sparse topology, which is not sufficient to explore the space well enough to find a reliable model". This is obviously false even for the one sparse training method they use, RiGL. The idea of dynamic sparse training (DST) methods not exploring the space well is explored in "Do We Actually Need Dense Over-Parameterization? In-Time Over-Parameterization in Sparse Training" by Liu et al., and there it's shown that DST methods can be made to explore most of the parameter space using the ITOP measure.
* When motivating the CiGL method in 3.1 the authors claim that "...added randomness in the weight update step and leads to better exploration of the weight space.". Better than what? Assuming RiGL/other sparse training methods, this statement has no evidence behind it, and could be quantified as in existing work, i.e. with ITOP. The authors could instead show the difference in ITOP for RiGL and CiGL to present an analysis that would quantify the difference in exploration.

**Summary Of The Paper:**

While it is known that Deep Neural Networks (DNNs) are in over-confident, and require calibration in practice, the authors empirically show that a popular dyanamic sparse training method, Rigging the Lottery Ticket (RiGL) appears to suffer from worse confidence calibration than a comparable dense model in practice. The authors propose a method based on RiGL, The Calibrated Lottery Ticket (CiGL), to improve confidence calibration while achieving similar or better generalization. The authors provide a proof claiming to show that their proposed CiGL method can be considered a hierarchical variational approximation to a probabilistic deep Gaussian process. The authors demonstrate the effectiveness of their method compared to standard RiGL, as well as RiGL calibrated with existing methods for dense models, on the ImageNet, CIFAR10, and CIFAR-100 datasets with a range of relevant models. The authors also perform an ablation analysis of CiGL to understand the contributions of different aspects of the method.

**Summary Of The Review:**

This paper presents a novel and pressing finding that will be of great interest to the sparse training research community. Overall it's well written with a good empirical evaluation, and motivation. Unfortunately the misleading over-claim of the comprehensiveness of the empirical evaluation significantly weakens what is otherwise a very strong contribution. There are also potentially significant issues with the algorithm presented and the paper's clarity on some points. Overall *most* of these issues are relatively easy to address, and the paper has the potential to be a strong contribution that I would be happy to see accepted if they are in the rebuttal (and my rating below at present reflects the assumption they will be).

---

> ### Author Response · Authors · 2022-11-13
> **Response to reviewer db8U (Q1-Q3)**
>
> Thank you for the insightful comments and for acknowledging the value of our work. Please find our responses to the questions as follows.
>
> **Q1: The paper only looks at one sparse training method: RiGL (not even SET). As far as I'm aware, the authors are the first to identify and study the problem of calibration in sparse training, and this would be a much better supported (and very strong) claim to make already.**
>
> A1: Thank you for the suggestion. We have updated our statement to "first to identify and study the problem of calibration in sparse training".
>
> RigL is the most popular and typical sparse training method, and essentially it enjoys the same paradigm and computational framework as other sparse training methods. Therefore, the conclusions about RigL can be generalized to most other sparse training methods.
>
> In addition, we add some experiments using SET and find that the sparse model produced by SET is also more over-confident than the dense model. As shown in the table below, the ECE values of dense ResNet-50 are smaller than those of sparse ResNet-50 on both CIFAR-100 and CIFAR-10.
>
> | Sparsity |  0%  |  50%  |  80%  |  90%  |  95%  |  99%  |
> | --------- | ----- | ----- | ----- | ----- | ----- | ----- |
> | CIFAR-100 (ResNet-50) | 0.0841 | 0.0931 | 0.1058 | 0.1290 | 0.1282 | 0.0873 |
> | CIFAR-10 (ResNet-50) | 0.0381 | 0.0429 | 0.0416 | 0.0459 | 0.0460 | 0.0589 |
>
> **Q2: In Algorithm 1, the specific pseudocode used for updating the joint weight/mask running mean is known for being an extremely numerically unstable method in practice to calculate a running mean, and will for any moderate number of time steps result in a significant loss in floating point precision.**
>
> A2:  In our work, we repeat all the experiments 3 times and report the mean value and standard deviation. We do not use numerically stable algorithms yet. We will use a more stable algorithm in our future study, such as Welford’s algorithm.
>
> We think the reason it is less sensitive to the expected loss in precision is that we start to collect models and calculate the running mean near the end of training. At this stage, the models are close to convergence and all of them are located in the same low-error basin. As a result, the differences between the different model weights are not too large, leading to reduced sensitivity. It was also mentioned in [R1] that the collected model samples were located in the same error basin.
>
> In addition, many works related to weight averaging have used this formula and shown good performance [R1, R2].
>
> [R1] Izmailov, Pavel, et al. "Averaging weights leads to wider optima and better generalization." arXiv preprint arXiv:1803.05407 (2018).
>
> [R2] Nikishin, Evgenii, et al. "Improving stability in deep reinforcement learning with weight averaging." Uncertainty in artificial intelligence workshop on uncertainty in Deep learning. 2018.
>
>
> **Q3: The idea of dynamic sparse training (DST) methods not exploring the space well is explored in "Do We Actually Need Dense Over-Parameterization?**
>
> A3: Our work has a different goal from ITOP with respect to encouraging exploration of the weight space, and also addresses a limitation of ITOP. As for the exploration of weight space, there are two aspects, namely “which weight is active” & “what value that weight has”.
>
> (i) Difference: while our work aims to better explore the weight space to find more reliable models, the goal of ITOP is to build models with higher accuracy.
>
> (ii) Limitation of ITOP: given the mask, the second aspect is not addressed and the optimization of the algorithm remains more challenging than dense training due to the pseudo-local optimization introduced by the sparsity constraints. To meet this challenge, ITOP increases the iterations between mask updates, leading to an increase in training time.
>
> We illustrate how our work addresses this limitation.
>
> - The first aspect of weight space exploration: It is reflected by the ITOP rate, which is the percentage of all weights that have ever been selected as active weights by the mask.
> - The second aspect of weight space exploration: It is reflected by the idea of "reliable exploration" in the ITOP paper. Ideally, a reliable exploration should allow a model to find the good direction and jump out of the bad local optimum.
> - The sparsity constraints introduce some pseudo-local optima, which is difficult to jump out of. Our random mask can randomly cut off some directions and force the model to explore other directions, thus encouraging the model to better explore the weight space and avoid missing the correct direction.
>
> We have updated our statement and included more discussions of ITOP in the new manuscript.

---

> > ### Author Response · Authors · 2022-11-13
> > **Response to reviewer db8U (Q4-Q9)**
> >
> > **Q4: When motivating the CiGL method in 3.1 the authors claim that "...added randomness in the weight update step and leads to a better exploration of the weight space.". Better than what? The authors could instead show the difference in ITOP for RiGL and CiGL to present an analysis that would quantify the difference in exploration.**
> >
> > A4: Our CigL leads to a better exploration of the weight space compared to RigL.
> >
> > As described in A3, our CigL aims to encourage more exploration of weight values (the second aspect), while the ITOP rate reflects the exploration of which weight is active (the first aspect). Therefore, the ITOP rate cannot quantify the advantage of our CigL in weight space exploration.
> >
> > The reasons why our CigL can encourage better exploration of weight values are explained below.
> >
> > - The difficulty in exploring the weight values is due to the sparsity constraint, which cuts off the routes and creates some pseudo-local optima [R3, R4, R5].
> > - In sparse training with only one mask, the model either takes a long time to jump out of the pseudo-local optimum or gets stuck in it.
> > - Our random mask can randomly cut off some directions and force the model to explore other directions that might not be explored without enforcement, thus encouraging the model to better explore the weight space and jump out of the bad local optimum.
> >
> > [R3] Evci, Utku, et al. "The difficulty of training sparse neural networks." arXiv preprint arXiv:1906.10732 (2019).
> >
> > [R4] Sun, Yiyou, and Yixuan Li. "On the Effectiveness of Sparsification for Detecting the Deep Unknowns." arXiv preprint arXiv:2111.09805 (2021).
> >
> > [R5] He, Zheng, et al. "Sparse Double Descent: Where Network Pruning Aggravates Overfitting." ICML, 2022.
> >
> > **Q5: ECE acronym is not defined until page 6**
> >
> > A5: Thank you for pointing out the lack of clarity regarding the ECE metric. We add a definition and more illustrations in the introduction.
> >
> > **Q6: ADOPT seems to be used instead of CiGL in a number of the figures/tables, without being explained. I'm assuming that it's just an old name for CiGL that stuck around...**
> >
> > A6: Yes, ADOPT is an old name and we have updated all of them to CigL in the new manuscript version.
> >
> > **Q7: There is much more focus on the existence of proof that there is an equivalence between CiGL and variational approximations of a deep Gaussian process itself than motivating why this should matter to the reader! Section 4.2. should be the introduction to 4.1 to address this, and in the introduction, a brief statement based on 4.2 should be added to motivate this contribution.**
> >
> > A7: We have updated section 4 as suggested to make the motivation clearer.
> >
> > **Q8: Multiple times it's claimed that both the model weights and masks are "averaged", and this is true, but the way this is written led me at least to believe these operations were independent, and in particular wonder what "mask averaging" is/can be affected. In Algorithm 1 a (joint) running average of the masked weights is performed, and it appears that this is what is being discussed. Although it's possible I'm in the minority in how I read this, I would reword this to avoid giving the reader the impression these are independent.**
> >
> > A8: Thank you for pointing out the lack of clarity on weight \& mask averaging. We have updated the description to make it clearer.
> >
> > In Algorithm 1, our weight & mask averaging can be viewed as an extension to the weight averaging method, where we use random masks for weighted averaging over the sparse weights. We collect pairs of sparse weights $\textbf{W}^{(t)}$ and random mask $\textbf{Z}^{(t)}$ at each epoch near the end of training. Then, we perform elemental multiplication of $\textbf{W}^{(t)}$ and $\textbf{Z}^{(t)}$ and take the average to obtain the final output sparse model.
> >
> > **Q9: I would mention, briefly, the connection with dropout earlier in the introduction if only to mention to the reader that it's discussed later in the paper.**
> >
> > A9: Thank you for the suggestion. We add a statement about the connection with dropout in the introduction.

---

> > > ### Comment · Reviewer_db8U · 2022-11-18
> > > **Thanks for rebuttal comments**
> > >
> > > First want to thank the authors for their rebuttal comments, and to acknowledge I have read them.
> > >
> > > I'm still overall happy with much of this paper, and most of the points that have been addressed by the authors with the exception of the numerically unstable method of calculating the running mean of weights.
> > >
> > > I should have been more clear that rather than being a small loss of precision, over even a moderate number of time steps this loss in precision can be catastrophic! Unfortunately given this I don't find the authors explanation to be much comfort, it would be much more convincing even over a small validation run to know what the actual loss in precision is v.s. a numerically stable method.

---

> > > > ### Author Response · Authors · 2022-11-19
> > > > **About the numerically unstable problem**
> > > >
> > > > Thank you very much for your reply and the valuable comment!
> > > >
> > > > As suggested by the reviewer, we add experiments in which we use the numerically stable method Welford's algorithm to calculate running averages in our weight & mask averaging process (Wide-ResNet-22-2, CIFAR-10).
> > > >
> > > > We compare CigL and CigL + Welford's algorithm in terms of (i) test accuracy, (ii) test loss, and (iii) ECE. We also plot the (iv) mean squared differences between the weights of CigL and CigL + Welford's algorithm. Since we started using weight and mask averaging in the 200th epoch, we report results every 5 epochs starting from the 200th epoch.
> > > >
> > > > The results are summarized below, and we find that CigL and CigL + Welford's algorithm provide very similar results.
> > > >
> > > > (i) Test accuracy comparison: as shown in the table below, the test accuracy of CigL and CigL + Welford's algorithm are always very similar.
> > > >
> > > > | Epoch |  200 |  205 |  210 |  215 |  220 |  225  | 230 |  235 |  240 |  245  |
> > > > | :---------: | -----: | -----: | -----: | -----: | -----: | -----: | -----: | -----: | -----: | -----: |
> > > > | CigL (90%)| 91.26 | 92.45 | 92.55 | 92.55 | 92.79 | 92.81 | 92.89 | 93.03 | 93.07 | 93.03|
> > > > | CigL + Welford  (90%)| 91.26 |92.24 |92.53 |92.63 |92.73 |92.81 |92.77 |93.02 |92.91 |92.90 |
> > > > | CigL (99%)| 82.83 |83.97 |83.98 |83.79 |84.18 |83.95 |84.18 |84.11 |84.19 |83.89 |
> > > > | CigL + Welford  (99%)| 82.83 |83.77 |84.0 |83.97 |84.17 |83.92 |83.96 |84.21 |84.10 |83.85 |
> > > >
> > > > (ii) Test loss comparison: as shown in the table below, the test loss of CigL and CigL + Welford's algorithm are always very similar.
> > > >
> > > > | Epoch |  200 |  205 |  210 |  215 |  220 |  225  | 230 |  235 |  240 |  245  |
> > > > | :---------: | -----: | -----: | -----: | -----: | -----: | -----: | -----: | -----: | -----: | -----: |
> > > > | CigL (90%)| 0.2621 |0.2275 |0.2181 |0.2182 |0.2149 |0.2165 |0.2127 |0.2118 |0.2121 |0.2117 |
> > > > | CigL + Welford  (90%)| 0.2621 |0.2306 |0.2181 |0.2226 |0.2158 |0.2194 |0.2170 |0.2120 |0.2148 |0.2182 |
> > > > | CigL (90%)| 0.5061 |0.4731 |0.4663 |0.4709 |0.4631 |0.4598 |0.4594 |0.4568 |0.455 |0.4601 |
> > > > | CigL + Welford  (90%)| 0.5061 |0.4772 |0.4656 |0.4704 |0.4637 |0.4641 |0.4657 |0.4567 |0.4599 |0.4609 |
> > > >
> > > > (iii) ECE comparison: as shown in the table below, the ECE of CigL and CigL + Welford's algorithm are always very similar.
> > > >
> > > > | Epoch |  200 |  205 |  210 |  215 |  220 |  225  | 230 |  235 |  240 |  245  |
> > > > | :---------: | -----: | -----: | -----: | -----: | -----: | -----: | -----: | -----: | -----: | -----: |
> > > > | CigL (90%)| 0.0159 |0.0123 |0.0147 |0.0143 |0.0134 |0.0135 |0.0146 |0.0131 |0.0137 |0.0139 |
> > > > | CigL + Welford  (90%)| 0.0159 |0.0121 |0.0149 |0.0144 |0.0131 |0.0131 |0.0145 |0.0131 |0.0134 |0.0140 |
> > > > | CigL (90%)| 0.0123 |0.0161 |0.0144 |0.0126 |0.0119 |0.0120 |0.0098 |0.0095 |0.0102 |0.0114 |
> > > > | CigL + Welford  (90%)| 0.0123 |0.0157 |0.014 |0.0130 |0.0122 |0.0123 |0.0096 |0.0089 |0.0104 |0.0113 |
> > > >
> > > > (iv) Mean square difference: as shown in the table below, although the mean square differences between the weights of CigL and CigL + Welford's algorithm increase with the number of epochs, the difference is less than 1e-12, which is very small.
> > > >
> > > > | Epoch |  200 |  205 |  210 |  215 |  220 |  225  | 230 |  235 |  240 |  245  |
> > > > | :---------: | -----: | -----: | -----: | -----: | -----: | -----: | -----: | -----: | -----: | -----: |
> > > > | Difference (90%)| 0.0e-14  |0.7194e-14  |1.4655e-14  |0.3020e-14  |1.5277e-14  |0.2309e-14  |0.9881e-14  |1.4477e-14  |1.3500e-14  |1.7497e-14  |
> > > > | Difference (99%)|0.0e-14  |0.9072e-14  |1.0404e-14  |3.1720e-14  |1.3169e-14  |2.1760e-14  |7.7895e-14  |22.7224e-14  |17.8555e-14  |2.3804e-14  |

---

> > > > > ### Comment · Reviewer_db8U · 2022-11-21
> > > > > **Rebuttal**
> > > > >
> > > > > Thanks - these results are definitely reassuring and suggest there are no numerical issues that are affecting the results for your situation considering it appears to be negligible over 200 epochs.

---

> > > > > > ### Author Response · Authors · 2022-11-22
> > > > > > **Thank you again!**
> > > > > >
> > > > > > Thank you again for your insightful comments and valuable suggestions!

---

### Official Review · Reviewer_JN9S · 2022-10-23

**Confidence:** 5
**Correctness:** 3
**Technical Novelty And Significance:** 2
**Empirical Novelty And Significance:** 2
**Recommendation:** 8

**Clarity, Quality, Novelty And Reproducibility:**

The paper is straightforward and quite clear to understand. However, the technique novelty is significantly insufficient given the overlap with prior works.

**Strength And Weaknesses:**

## Strength

(1) The paper is well-motivated and straightforward to read.

(2) The research topic is important for sparse training.

(3) The proof of CigL can be viewed as a hierarchical variational approximation has its theoretical values.

## Weaknesses

(1) The paper claims that they comprehensive study on the reliability of sparse training. However, they narrowly focus on only one sparse training method RigL, which is far away from a comprehensive study. Since gradient is used to grow new weights, it is natural for RigL to be over-confident. To draw a more general conclusion, we need to study other sparse training methods, at least SET (Mocanu2018) that randomly activates weights should be included. Do we also observe the same over-confident in other types of sparse training? For instance, it has been shown that Lottery Tickets enjoy better uncertainty estimation than dense networks [1,3].

(2) The paper claims that they are the first to study the reliability of sparse training, which I am afraid is not true. Without specifically searching, I can list at least three recent works on the reliability of sparse training [1,2,3] which are completely overlooked in the submisison. Specifically, I find the CigL algorithm shares a significant overlap with [2] where exactly the same weight averaging is used to improve the performance and reliability of sparse training. Besides, [2] has also studied the effectiveness of cyclic learning rates and other types of sparse network averaging. I highly encourage the author to distinguish CigL itself from these prior works. Maybe directly comparing with them to show CigL's superiority.

(3) Since prior works have shown the power of weight averaging [2] of subnetworks in improving performance, robustness, and uncertainty estimation, it is not surprising that CigL can outperform RigL in uncertainty estimation.

(4) I expect there should be a pruning operation after averaging of two subnetworks with different masks, to ensure the same sparsity level after averaging. Unless I misunderstand, could the authors clarify why CigL does not need this pruning operation?

(5) I understand that CigL can be viewed as an approximation to the deep GP using hierarchical variational inference after reading their proof. Could the authors elaborate more on how this approximation benefits sparse training further in terms of reliability?

(6) Many terms are not introduced properly, such as ADOPT and W_CigL. What architectures and datasets are used for Figure 1?

[1] Calibrate and Prune: Improving Reliability of Lottery Tickets Through Prediction Calibration. AAAI 2021

[2] Superposing Many Tickets into One: A Performance Booster for Sparse Neural Network Training. UAI 2022.

[3] Can You Win Everything with A Lottery Ticket? TMLR.

**Summary Of The Paper:**

This paper focuses on studying the reliability of sparse training. The authors specifically analyze the reliability (the expected calibration error) of RigL and find the solutions learned by RigL are often over-confident. Stemming from this observation, they propose to draw several random masks besides the one learned by RigL, and an average is used to perform a weight-level ensemble to calibrate sparse models. I believe this research topic is important for the sparse training community. However, this paper is overall narrowly targeted and completely misses several important prior works on this topic, significantly diluting its values.

**Summary Of The Review:**

This paper studies an interesting topic for sparse training. However, given the insufficient study of different sparse training methods and a significant overlap with prior works, the merits of this paper are degraded largely.


## After rebuttal

I am satisfied with the authors' feedback. I raise the score to 8. I would like to see the promised changes in the revision.

Regarding related work R1, R2 and R3, I understand the difference between LTH and sparse training. However, they should be discussed as related work, given their highly related research goals and topics to this paper (i.e., sparse NNs and calibration).

---

> ### Author Response · Authors · 2022-11-13
> **Response to reviewer JN9S (Q1-Q2)**
>
> Thank you for the review and for bringing up these questions. Please see our responses below to clarify the main concerns.
>
>
> **Q1: The paper narrowly focuses on only one sparse training method RigL, which is far away from a comprehensive study. At least SET (Mocanu2018) that randomly activates weights should be included. Do we also observe the same over-confident in other types of sparse training? For instance, it has been shown that Lottery Tickets enjoy better uncertainty estimation than dense networks [R1, R2].**
>
> A1: (i) We have updated our statement to "first to identify and study the problem of calibration in sparse training". As the reviewer db8U mentioned, this is still a very strong contribution.
>
> (ii) RigL is the most popular and typical sparse training method, and essentially it enjoys the same paradigm and computational framework as other sparse training methods. Therefore, the conclusions about RigL can be generalized to most other sparse training methods.
>
> (iii) As for SET, we add some experiments using SET and find that the sparse model produced by SET is also more over-confident than the dense model. As shown in the table below, the ECE values of dense ResNet-50 are smaller than those of sparse ResNet-50 on both CIFAR-100 and CIFAR-10. This proves the reliability issue of sparse training.
>
> | Sparsity |  0%  |  50%  |  80%  |  90%  |  95%  |  99%  |
> | --------- | ----- | ----- | ----- | ----- | ----- | ----- |
> | CIFAR-100 (ResNet-50) | 0.0841 | 0.0931 | 0.1058 | 0.1290 | 0.1282 | 0.0873 |
> | CIFAR-10 (ResNet-50) | 0.0381 | 0.0429 | 0.0416 | 0.0459 | 0.0460 | 0.0589 |
>
> (iv) As for Lottery Tickets [R1, R2], they belong to pruning methods and are different from sparse training. However, our work focuses on the reliability of sparse training. We include[R1, R2] as related work in the references.
>
> (v) In many cases, pruning can also be detrimental to confidence calibration. For example, in [R1], as shown in Figure 1 (c), the ECE value increases with the compression rate. For [R2], it is also mentioned that uncertainty measures are more sensitive to pruning than generalization metrics. As shown in Figures 5 and 6 of [R2], the sparse model can have a higher ECE value compared to the dense model at multiple sparsity levels.
>
> **Q2: The paper claims that they are the first to study the reliability of sparse training, which I am afraid is not true. I highly encourage the author to distinguish CigL itself from these prior works [R1, R2, R3]. Maybe directly comparing with them to show CigL's superiority.**
>
> A2: Our work is significantly different from [R1, R2, R3], which is illustrated below.
>
> (i) [R1] and [R2] belong to pruning methods which are different from sparse training. Our work is to study the reliability of sparse training and to design a method to solve the more pronounced over-confidence problem in sparse training.
>
> (ii) For [R3], it is a concurrent paper and we are not aware of this work. It was accepted by UAI 2022 which was held in August 2022 and the deadline for ICLR submission is September.
>
> (iii) Our work still differs from [R3] in several important aspects.
>
> - The goal of our CigL is different from [R3]. Our method is to improve the reliability of sparse training. In contrast, [R3] aims to boost the performance of sparse models, and most of the results are based on accuracy improvements. They only show the uncertainty estimation in a short subsection at the end of the paper.
> - Our CigL is much more effective in reducing ECE value and improving confidence calibration. We summarize the change in ECE values for sparse ResNet-50 on CIFAR10/100. As shown in the table below, [R3] has a limited reduction in ECE values, which is much smaller than our CigL.
>
> |  |   CIFAR-10: 80%    | CIFAR-10: 90% |   CIFAR-10: 95%   | |   CIFAR-100: 80%    | CIFAR-100: 90% |   CIFAR-100: 95%   |
> | :---------: | :-----: | :-----: | :-----: | :- | :-----: | :-----: | :-----: |
> |    R3    | -0.0012 | -0.0005 | -0.0007 | | -0.0005 | -0.0010 | -0.0010 |
> |   CigL   | **-0.0067** | **-0.0080** | **-0.0119** | | **-0.0141** | **-0.0113** | **-0.0104** |
>
> - Our CigL has theoretical justifications and can be viewed as a hierarchical variational approximation to the deep Gaussian process, which guarantees a better confidence calibration.
> - Our CigL method is different from [R3] and does not require further pruning steps. However, [R3] requires further pruning after averaging, which adds additional cost.
> - In our work, we investigate how sparse training will affect the reliability of the model. But [R3] does not consider and study this issue.
>
> [R1] Calibrate and Prune: Improving Reliability of Lottery Tickets Through Prediction Calibration. AAAI 2021.
>
> [R3] Can You Win Everything with A Lottery Ticket? TMLR.
>
> [R3] Superposing Many Tickets into One: A Performance Booster for Sparse Neural Network Training. UAI 2022.

---

> > ### Author Response · Authors · 2022-11-13
> > **Response to reviewer JN9S (Q3-Q4)**
> >
> > **Q3: Since prior works have shown the power of weight averaging [R3] of subnetworks in improving performance, robustness, and uncertainty estimation, it is not surprising that CigL can outperform RigL in uncertainty estimation.**
> >
> > A3: We politely disagree with the reviewer's view. Both random mask (RM) and weight & mask averaging (WMA) are important and indispensable to improving confidence calibration. Weight averaging alone can improve performance, but is ineffective in improving uncertainty estimation.
> >
> > (i) Empirically, we have done ablation studies in Section 5.4 to show the effect of WMA. We find that when only WMA is used (CigL w/o RM), the ECE value cannot be effectively reduced. As shown in Figure 5 (b), the blue, green, and pink columns represent RigL, CigL w/o RM, and CigL (ours). The ECE value increases at sparsity 80\% and remains almost constant at sparsity 99\%. For sparsity 90\% and 95\%, it still does not decrease as much as when using the RM and WMA together.
> >
> > We summarize the changes in ECE values when using our CigL and when using only the WMA (CigL w/o RM). As shown in the table below, CigL is usually effective in reducing ECE values, but CigL w/o RM either increases ECE values or has only a limited reduction.
> >
> > | Sparsity |  80%  |  90%  |  95%  |  99%  |
> > | :---------: | -----: | -----: | -----: | -----: |
> > | CigL w/o RM (cifar-100, ResNet-50)| 0.0038 | -0.0098 | -0.0144 | -0.0046 |
> > | CigL (cifar-100, ResNet-50) | **-0.0010** | **-0.0152** | **-0.0344** | **-0.0269** |
> > | CigL w/o RM (cifar-10, Wide-ResNet-22-2) | 0.0060 | 0.0129 | 0.0149 | 0.0017 |
> > | CigL (cifar-10, Wide-ResNet-22-2) | **-0.0141** | **-0.0113** | **-0.0104** | **-0.0049**|
> >
> > (ii) Empirically, as shown in the Table below, [R3] has a limited reduction in the ECE values, which is much smaller than our CigL.
> > |  |   CIFAR-10: 80%    | CIFAR-10: 90% |   CIFAR-10: 95%   | |   CIFAR-100: 80%    | CIFAR-100: 90% |   CIFAR-100: 95%   |
> > | :---------: | :-----: | :-----: | :-----: | :- | :-----: | :-----: | :-----: |
> > |    R3    | -0.0012 | -0.0005 | -0.0007 | | -0.0005 | -0.0010 | -0.0010 |
> > |   CigL   | **-0.0067** | **-0.0080** | **-0.0119** | | **-0.0141** | **-0.0113** | **-0.0104** |
> >
> > (iii) Existing work also supports our claim. When only using WMA without a random mask, the averaging procedure is equivalent to weight averaging. As mentioned in [R4], although weight averaging can produce better generalization, it does not improve confidence calibration. This suggests that the improvement in confidence calibration comes not only from the WMA but also from the RM.
> >
> > [R3] Superposing Many Tickets into One: A Performance Booster for Sparse Neural Network Training. UAI 2022.
> >
> > [R4] Wortsman, Mitchell, et al. "Model soups: averaging weights of multiple fine-tuned models improves accuracy without increasing inference time." International Conference on Machine Learning. PMLR, 2022.
> >
> > **Q4: I expect there should be a pruning operation after averaging two subnetworks with different masks, to ensure the same sparsity level after averaging. Unless I misunderstand, could the authors clarify why CigL does not need this pruning operation?**
> >
> > A4: Our CigL does not require additional pruning operations after averaging two subnetworks, which can save more cost compared to [R3]. The reasons for this are illustrated below.
> >
> > - Suppose we train a network with 95\% sparsity. The deterministic mask $\textbf{M}$ will have 95\% sparsity.
> > - The random mask $\textbf{Z}^{(t)}$ (like 10\% sparsity) further deactivates 10\% active weights to produce subnetwork at 95.5\% sparsity.
> > - We collect sparse models and random masks near the end of training. At this point, the deterministic mask $\textbf{M}$ will be fixed due to the decay of the pruning rate like RigL, and only the random mask $\textbf{Z}^{(t)}$ will be updated.
> > - Since $\textbf{Z}^{(t)}$ only cuts off the connections and does not add new connections, the set of active weights after averaging will be within the active weights chosen by $\textbf{M}$.
> > - Thus, we will get a model with at least 95\% sparsity whose sparse topology is controlled by $\textbf{M}$.
> >
> > In summary, as long as $\textbf{M}$ is fixed in the collected model and $\textbf{Z}^{(t)}$ only deactivates the weights, we do not need additional pruning operations.

---

> > > ### Author Response · Authors · 2022-11-13
> > > **Response to reviewer JN9S (Q5-Q6)**
> > >
> > > **Q5: I understand that CigL can be viewed as an approximation to the deep GP using hierarchical variational inference after reading their proof. Could the authors elaborate more on how this approximation benefits sparse training further in terms of reliability?**
> > >
> > > A5: (ii) In CigL, the hierarchical structure can help to expand the family of variational distributions, leading to a better approximation of the posterior. Thus, it enables the Bayesian method to show its power and leads to improved confidence calibration in sparse training.
> > >
> > > (i) Bayesian methods have shown the ability to improve confidence calibration. However, in sparse training, it becomes more challenging to approximate the posterior well, so Bayesian methods cannot show their full advantage in improving confidence calibration. The hierarchical Bayesian approximation method (CigL) addresses this challenge.
> > >
> > > **Q6: Many terms are not introduced properly, such as ADOPT and $\textbf{W}_{CigL}$. What architectures and datasets are used for Figure 1?**
> > >
> > > A6: We have corrected these problems in the new manuscript version.
> > > - ADOPT is a prior name for the method and we have updated it to CigL in the new manuscript version.
> > > - For $\textbf{W}_{CigL}$, it is the final sparse model weights produced by our CigL.
> > > - For Figure 1, we are using ResNet-50 on CIFAR-100.

---

> > > > ### Comment · Reviewer_JN9S · 2022-11-21
> > > > **Thanks for the clarification**
> > > >
> > > > I thank the authors for their explanation, which has addressed some of my concerns. It is good to know that only averaging leads to marginal uncertainty improvement and no pruning operation is required after weight averaging. Therefore, I increased my score to 5 (not 6 because there still remain some concerns that are not addressed).
> > > >
> > > > * I appreciate the authors explaining the differences between this paper and R1, R2, and R3. I would like to see them in the early part of the paper, not the appendix. Moreover, I suggest the authors remove the misleading claim that "Existing sparse training methods use only one mask to determine the sparse topology, which can suffer from spurious local minima and make it difficult to find a reliable model." At least in previous work R3, using multi masks to chase better accuracy and uncertainty has already been probed. I believe there are more if we do a search in the related work.
> > > >
> > > > * The Authors have missed three very closely related works (R1, R2, and R3), which in my opinion cannot be fully justified by R1 and R2 are Lottery Tickets related, and R3 being published close to ICLR submission. IMHO, Lottery Tickets and sparse training are highly related, since their ultimate goals are all discovering trainable subnetworks that can be trained from scratch to high accuracy, although their discovering journeys are different. While R3 was presented in UAI in August 2022, it has been on arxiv for a long time (this is why I can recall it).
> > > >
> > > > * While the RigL is the most popular sparse training method (which is also very arguable), it is apparently not scientific and convincing enough that we draw our conclusions only based on one method. From the ablation study of weight averaging (WA) and random weight, I can see that WA itself is insufficient to improve uncertainty and random weight (RM) is more vital. However, I can not understand why SET also suffers from poor uncertainty since the random mask operation is inherently involved in SET. Does this in turn show that random exploration isn't sufficient to guarantee good uncertainty performance? I find this explanation quite confusing.
> > > >
> > > > * Can the improvement be achieved on more challenging tasks like ImageNet? This is important since the masking operation naturally increases the sparsity, which would be fine on simple datasets like CIFAR, but might hurt the performance on Imagetnet.
> > > >
> > > > [R1] Calibrate and Prune: Improving Reliability of Lottery Tickets Through Prediction Calibration. AAAI 2021.
> > > >
> > > > [R3] Can You Win Everything with A Lottery Ticket? TMLR.
> > > >
> > > > [R3] Superposing Many Tickets into One: A Performance Booster for Sparse Neural Network Training. UAI 2022.

---

> > > > > ### Author Response · Authors · 2022-11-23
> > > > > **Response to the remaining concerns (Q1-Q2)**
> > > > >
> > > > > Thank you very much for your reply and valuable comments! Please see our responses below to clarify the remaining concerns.
> > > > >
> > > > > **Q1: I appreciate the authors explaining the differences between this paper and R1, R2, and R3. I would like to see them in the early part of the paper, not the appendix. Moreover, I suggest the authors remove the misleading claim that "Existing sparse training methods use only one mask to determine the sparse topology, which can suffer from spurious local minima and make it difficult to find a reliable model." I believe there are more if we do a search in the related work.**
> > > > >
> > > > > A1: As suggested by the reviewers,
> > > > >
> > > > > (i) We have moved [R1], [R2], and [R3] to the related work in Section 2 and have added detailed explanation of the differences. We have also removed the misleading statement.
> > > > >
> > > > > (ii) We have done a more comprehensive literature review of related work using multiple masks and updated Section 2 [R4, R5].
> > > > > - [R4] considers sparse training in federated learning and investigates the aggregation of multiple masks in edge devices.
> > > > > - [R5] utilizes multiple masks with different granularities to allow greater flexibility in structured pruning and to improve accuracy.
> > > > > - Despite the use of multiple masks, [R4, R5] differ significantly from our work. They still use deterministic masks, which still suffer from the lack of exploration of the weight space, and consider only the accuracy of sparse models. In addition, their setups are federated learning and pruning, which are different from our work.
> > > > >
> > > > > Since we cannot upload the new version at this discussion stage, these changes will be shown in the final version.
> > > > >
> > > > > **Q2: The Authors have missed three very closely related works (R1, R2, and R3), which in my opinion cannot be fully justified by R1 and R2 are Lottery Tickets related, and R3 being published close to ICLR submission. IMHO, Lottery Tickets and sparse training are highly related, since their ultimate goals are all discovering trainable subnetworks that can be trained from scratch to high accuracy, although their discovering journeys are different. While R3 was presented in UAI in August 2022, it has been on arxiv for a long time.**
> > > > >
> > > > > A2.1 (**Comparison with [R1, R2]**):
> > > > >
> > > > > For [R1] and [R2], although Lottery Tickets and sparse training are related and both produce subnetworks with high accuracy, their goals and discovering journeys are quite different, which leads to significant differences in several important properties, including uncertainty, geometry of the loss surface, generalization ability, and so on.
> > > > >
> > > > > (i) For the goal, Lottery Tickets mainly aim to reduce the inference cost, while the sparse training also aims to save resources during the training phase.
> > > > >
> > > > > (ii) Lottery Tickets and sparse training are different in several important properties. As shown in Figure 11 of [R2], the blue and purple bars represent Lottery Tickets and sparse training with a sparsity level of 79%, respectively.
> > > > > - For uncertainty, sparse training does not improve the confidence calibration compared to dense training, while Lottery Tickets allows for improved confidence calibration.
> > > > > - For the geometry of the loss surface, sparse training leads to larger trace values and cannot locate flat local minima. In contrast, Lottery Tickets can still locate flat local minima.
> > > > > - For generalization ability, sparse training provides higher accuracy and improved robustness compared to dense training, while Lottery Tickets provide relatively less improvement.
> > > > >
> > > > > (iii) The main reason for the different properties is their different discovering journeys.
> > > > >
> > > > > For Lottery Tickets:
> > > > > - It retrains the weights from the initial training phase after each pruning, which significantly increases the training time but allows more time for the model to explore the weight space.
> > > > > - It starts from a dense model and has low sparsity in the early stages, which reduces the difficulty of weight space exploration caused by the sparsity constraints.
> > > > >
> > > > > For spare training:
> > > > > - It maintains a high level of sparsity throughout the training process, which does not extend the training time to enable more exploration of the weight space.
> > > > > - In addition, maintaining high sparsity can cut off a large portion of the optimization route and produce more spurious local minima, thus making training very difficult.
> > > > > - [R2] shows the differences in the properties of Lottery Tickets and sparse training at the 79% sparsity level. The difficulty of training typically increases with increasing sparsity, implying that the difference is likely to be greater at higher sparsity levels.
> > > > >
> > > > > In summary, Lottery Tickets and the sparse training are very different in many aspects. In contrast to [R1] and [R2], our work focuses on sparse training, which reveals the calibration issue in the sparse training, offers a simple and effective solution, and provides theoretical analysis and extensive empirical demonstration.
> > > > >
> > > > > The **comparison with [R3]** will be shown in A2.2 below.

---

> > > > > > ### Author Response · Authors · 2022-11-23
> > > > > > **Response to the remaining concerns (Q2-Q4)**
> > > > > >
> > > > > > A2.2 (**Comparison with [R3]**):
> > > > > >
> > > > > > For [R3], it is still very different from what we have done regardless of the releasing time.
> > > > > > - The goal of our work is different from [R3]. We study how sparse training will affect the reliability of the model and propose CigL to improve the reliability of sparse training. In contrast, [R3] aims to boost the performance of sparse models, and most of the results are based on accuracy improvements. They only show the uncertainty estimation in a short subsection at the end of the paper.
> > > > > > - Our CigL is much more effective in reducing ECE value and improving confidence calibration. We summarize the change in ECE values for sparse ResNet-50 on CIFAR10/100. As shown in the table below, [R3] has a limited reduction in ECE values, which is much smaller than our CigL.
> > > > > >
> > > > > > |  |   CIFAR-10: 80%    | CIFAR-10: 90% |   CIFAR-10: 95%   | |   CIFAR-100: 80%    | CIFAR-100: 90% |   CIFAR-100: 95%   |
> > > > > > | :---------: | :-----: | :-----: | :-----: | :- | :-----: | :-----: | :-----: |
> > > > > > |    R3    | -0.0012 | -0.0005 | -0.0007 | | -0.0005 | -0.0010 | -0.0010 |
> > > > > > |   CigL   | **-0.0067** | **-0.0080** | **-0.0119** | | **-0.0141** | **-0.0113** | **-0.0104** |
> > > > > >
> > > > > > - Our CigL has theoretical justifications and can be viewed as a hierarchical variational approximation to the deep Gaussian process, whereas [R3] does not include theoretical analysis.
> > > > > > - The methodology of our CigL is different from [R3] and does not require further pruning steps. In contrast, [R3] requires further pruning after averaging, which adds additional cost.
> > > > > >
> > > > > > **Q3: However, I can not understand why SET also suffers from poor uncertainty since the random mask operation is inherently involved in SET. Does this in turn show that random exploration isn't sufficient to guarantee good uncertainty performance? I find this explanation quite confusing.**
> > > > > >
> > > > > > Yes, random exploration in SET is not sufficient to guarantee a good performance of uncertainty estimation. The reason is that the exploration of the weight space consists of two aspects, namely (i) “which weight is active” and (ii) “what the weight value is”, while the random exploration only addresses the former (i) and the latter (ii) remains unresolved.
> > > > > >
> > > > > > (i) In SET, random exploration can add more randomness to the mask search, improving the first aspect of weight space exploration. Since it chooses weight growth randomly and does not utilize gradient information, it can encourage mask exploration and is less likely to be stuck with suboptimal masks.
> > > > > >
> > > > > > (ii) However, SET still cannot solve insufficient exploration in terms of the second aspect.
> > > > > > - Given the mask, the optimization of the algorithm remains more challenging than dense training. The sparsity constraints cut off a large portion of the optimization route and produce more pseudo local minima, which is difficult to jump out of. Therefore, the model is more likely to stuck in these pseudo minima and unable to have sufficient exploration of the weight value space.
> > > > > >
> > > > > > In contrast, our CigL can address the second aspect of lack of exploration.
> > > > > > - In CigL, the random mask randomly cuts off some directions, forcing the model to explore other directions that might not have been explored in the absence of enforcement, thus encouraging the model to better explore the weight space and jump out of the bad local optimum. This is similar to the reason that Dropout helps find a better optimum [R6]. As a result, our CigL is able to find sparse models with better uncertainty performance.
> > > > > >
> > > > > > **Q4: Can the improvement be achieved on more challenging tasks like ImageNet? This is important since the masking operation naturally increases the sparsity, which would be fine on simple datasets like CIFAR, but might hurt the performance on Imagetnet.**
> > > > > >
> > > > > > Yes, improvements can be achieved on more challenging tasks like ImageNet. As shown in Figure 3 at Section 5.1, the blue and pink bars represent RigL and our CigL, respectively.
> > > > > > - In Figure 3 (a), the pink bars are usually shorter than the blue bars, indicating improved confidence calibration from CigL on ImageNet.
> > > > > > - In Figure 3 (b), the pink bars have similar heights compared to the blue bars, which suggests that our CigL can maintain comparable accuracy on ImageNet.
> > > > > >
> > > > > > [R1] Calibrate and Prune: Improving Reliability of Lottery Tickets Through Prediction Calibration. AAAI 2021.
> > > > > >
> > > > > > [R2] Can You Win Everything with A Lottery Ticket? TMLR.
> > > > > >
> > > > > > [R3] Superposing Many Tickets into One: A Performance Booster for Sparse Neural Network Training. UAI 2022.
> > > > > >
> > > > > > [R4] Bibikar, Sameer, et al. "Federated dynamic sparse training: Computing less, communicating less, yet learning better." AAAI 2022.
> > > > > >
> > > > > > [R5] Xia, Mengzhou, Zexuan Zhong, and Danqi Chen. "Structured pruning learns compact and accurate models." arXiv preprint arXiv:2204.00408 (2022).
> > > > > >
> > > > > > [R6] Srivastava, Nitish, et al. "Dropout: a simple way to prevent neural networks from overfitting." JMLR (2014).

---

> ### Author Response · Authors · 2022-11-17
> **Thank you again for your review and hope you may take a quick look at our responses.**
>
> Dear Reviewer,
>
> Thank you very much for taking your valuable time to review our paper. We are fortunate to have received very informative feedback from your comments, which is indispensable for us to polish the paper into a better version.
>
> We were wondering **if you could take a few minutes to take a quick look at our responses** and **let us know if you have any remaining questions or concerns so that we can address them before the deadline**. Much appreciate!
>
> Also, if you feel that your original concerns have been resolved, we would appreciate it if you would update your evaluation to reflect this. Thank you!
>
> Sincerely,
>
> Authors

---

> ### Author Response · Authors · 2022-12-03
> **Thank you again!**
>
> Thank you again for your valuable comments and suggestions!
>
> In the final version, we will include more discussion on sparse NNs and calibration [R1, R2, R3] in the related work section, and also incorporate other promised changes.

---

### Official Review · Reviewer_HnVS · 2022-10-25

**Confidence:** 4
**Correctness:** 3
**Technical Novelty And Significance:** 4
**Empirical Novelty And Significance:** 4
**Recommendation:** 6

**Clarity, Quality, Novelty And Reproducibility:**

The writing is clear, aside from the issues discussed above. The quality of the work is high and I believe the study and proposed algorithm are novel. I'd like for the description of the algorithm to be improved such that I can more accurately compare the proposed algorithm to RigL and other sparse training algorithms.

**Strength And Weaknesses:**

The paper is well written and the method is clearly motivated by the studies on over confidence in RigL-trained models. The experimental results are encouraging for CigL as a method for producing high quality and well-calibrated sparse neural networks.

In some places I found the paper to be difficult to understand. The ECE metric isn't defined until section 5.1, but is referenced repeatedly in the abstract and introduction. I had to re-read the first few sections to understand whether lower or higher ECE was indicative of over confidence. I also found the details algorithm to be unclear, particularly how "M" is used and which weights receive gradients on each iteration. In Algorithm 1, I see no reference to how the "M" mask is applied to the weights. I do see the use of the "Z" mask during gradient computation, but without the application of "M" it would seem that the entire weight matrix must be maintained across training. This is in contrast to RigL, which is designed to enable sparse storage and computation throughout training (outside of periodic gradient computation for mask updates). I'd like to understand the answer to these questions to more accurately evaluate this paper and to explain why CigL appears to produce significantly higher quality models for a given level of sparsity.

Lastly, the phrase "RigL + ADOPT" is used in various figures but ADOPT is never defined. It seems this maybe a prior name for the proposed CigL technique?

**Summary Of The Paper:**

The authors study the calibration of models trained using the sparse training algorithm RigL and show that sparse models are more overconfident than dense architectures of similar accuracy. They propose CigL, a modified version of RigL intended to calibrate sparse models and demonstrate that it produces models that are equal to or higher accuracy than RigL while also having lower degrees of over confidence.

**Summary Of The Review:**

The paper is interesting and potentially impactful, but some information is missing or unclear which makes it challenging to evaluate the proposed algorithm relative prior work.

---

> ### Author Response · Authors · 2022-11-13
> **Response to reviewer HnVS (Q1-Q3)**
>
> Thank you for your valuable review. Please see our responses below to clarify the main concerns.
>
> **Q1: The ECE metric isn't defined until section 5.1, but is referenced repeatedly in the abstract and introduction.**
>
> A1: In order to make the definition of ECE clearer, we have added a definition and more illustrations in the introduction of the new version.
>
> ECE [R1] is a popular measure of the discrepancy between a model's confidence and true accuracy, with a lower ECE indicating better confidence calibration and higher reliability. Regarding the calculation, we divide [0, 1] into $K$ bins, including $B_1,\cdots,B_{K}$, and ECE is defined as
> $$
> \text{ECE} = \sum_{k=1}^K \frac{|B_k|}{n}|acc(B_k) - conf(B_k)|,
> $$
> where $acc(B_k)$ and $ conf(B_k)$ are average accuracy and confidence of bins $B_k$. We can see that the ECE takes a weighted average of the absolute differences between accuracy and confidence.
>
> [R1] Guo, Chuan, et al. "On calibration of modern neural networks." ICML, 2017.
>
> **Q2: I also found the details algorithm to be unclear, particularly how "$\textbf{M}$" is used and which weights receive gradients on each iteration.**
>
> A2: (i) We have updated the description of $\\textbf{M}$ in the new manuscript version to make it clearer. When some elements of $\\textbf{M}$ are 0, the corresponding weights are deactivated. The inactive weights will be set as zero and won't be updated during the training. The weight update equation is as below, which is in similar way to the single mask in RigL:
> $$
> \textbf{W}^{(t)} = \textbf{W}^{(t-1)} - \alpha_t \textbf{M} \odot \textbf{Z}^{(t)} \odot \nabla L(\textbf{M} \odot \textbf{Z}^{(t)} \odot \textbf{W}^{(t-1)}; \textbf{B}_t)
> $$
>
> (ii) Since $\textbf{M}$ is aligned with the single sparse mask in RigL, we do not need to maintain the entire weight matrix, and only store the active weights. Thus, it enables sparse storage and computation like RigL.
>
> (iii) In addition, as shown in the weight update equation, our random mask $\textbf{Z}$ will bring a composite mask $\textbf{M} \odot \textbf{Z}^{(t)}$ with higher sparsity, further reducing the number of active weights and saving more storage and computational costs compared to one $\textbf{M}$ in RigL.
>
> **Q3: ADOPT is never defined. It seems this may be a prior name for the proposed CigL technique?**
>
> A3: Thank you for pointing out this typo. It is a prior name for the method and we have updated it to CigL in the new manuscript version.

---

> > ### Comment · Reviewer_HnVS · 2022-12-12
> > **Reviewer Response**
> >
> > Thank you for your responses. In particular, your clarification of the weight update equation with the M factor was very helpful to understand the algorithm. Given these clarifications I'm increasing my score from 5 to 6, as I am more confident in my understanding of the algorithm and the results the authors present.

---

> > > ### Author Response · Authors · 2022-12-12
> > > **Thank you again!**
> > >
> > > Thank you again for your valuable comments and questions!

---

> ### Author Response · Authors · 2022-11-17
> **Thank you again for your review and hope you may take a quick look at our responses.**
>
> Dear Reviewer,
>
> Thank you very much for taking your valuable time to review our paper. We are fortunate to have received very informative feedback from your comments, which is indispensable for us to polish the paper into a better version.
>
> We were wondering **if you could take a few minutes to take a quick look at our responses** and **let us know if you have any remaining questions or concerns so that we can address them before the deadline**. Much appreciate!
>
> Also, if you feel that your original concerns have been resolved, we would appreciate it if you would update your evaluation to reflect this. Thank you!
>
> Sincerely,
>
> Authors

---

### Official Review · Reviewer_3t4S · 2022-10-26

**Confidence:** 4
**Correctness:** 2
**Technical Novelty And Significance:** 3
**Empirical Novelty And Significance:** 2
**Recommendation:** 6

**Clarity, Quality, Novelty And Reproducibility:**

The writing is mostly clear. The idea of combining the deterministic and random masks seems novel. The code is provided for reproducibility.

**Strength And Weaknesses:**

+The mechanism of random mask and deterministic mask is interesting, and there is also a connection to Gaussian Process and hierarchical variational inference.

+The empirical performance looks good.

-It seems that the weight moving averaging (WMA) process is not integrated into the analysis of section 4.1. On the other hand, the figure. 5 shows that WMA significantly reduces the ECE value, which is much more obvious than the random mask. With this evidence, the analysis of section 4.2 seems questionable since the major confidence calibration comes from WMA instead of the Bayesian formulation.

-The process in Algorithm.1 and the Bayesian approximation introduced in section 4.1 is not well aligned, especially when learning the mask $M$. In Algorithm.1, the mask $M$ is updated using weights and gradients, like RigL in every $\Delta T$ steps. In section 4.1, the mask $M$ is only related to the weight magnitude. Also, weight $W$ and $M$ are updated iteratively. Why the approximation of section 4.1 still holds when the learning of $M$ is so different? Also, WMA is omitted in section 4.1.

-In Eq.(3), the authors show that $q(M_l | U_l) \propto \text{exp} (M_l\odot |U_l|)$. This formulation seems wired: $M_l$ appeared in both left and right head sizes of $\propto$. In addition, it's confusing whether $M_l$ is learned or sampled from a distribution. From the context, authors show that $M$ is updated by maximizing $q(M_l | U_l)$. If $M$ is updated and has exact 0 or 1 values, why are you sampling $M$? In addition, if $M$ is sampled, this is still not aligned with Algorithm.1. Algorithm.1 updates $M$ deterministically.

-In the final section, the authors provide four arguments in (a), (b), (c), and (d), and they are not very well supported. For example, in (b), the authors say that "It gradually becomes equivalent to a shallow model." How this conclusion arrives? There are no experiments to show measurements like effective depth.

-What is ADOPT? This is not explained in the paper.


**Summary Of The Paper:**

In this paper, they proposed a new sparse training method to produce sparse models with improved confidence calibration. A deterministic mask and a random mask are introduced for exploiting the weight magnitude and gradients and exploration.

**Summary Of The Review:**

In summary, the idea of combining the deterministic and random masks seems novel. It's a good attempt to connect the proposed method with GP using hierarchical variational inference. However, there are some inconsistencies between the analysis of section 4.1 and Algorithm.1; for example, WMA is ignored, and the learning of $M$ is different. In addition, the confidence calibration seems to come from WMA instead of the Bayesian approximation. In addition, the arguments in section 6 are not well supported.

_____________________________________

The authors clarify most of my concerns. As a result, I increased my score to 6.

---

> ### Author Response · Authors · 2022-11-13
> **Response to reviewer 3t4S (Q1-Q3)**
>
> Thank you for your valuable review. Please see our responses below to clarify the main concerns.
>
> **Q1: The weight & mask averaging (WMA) process is not integrated into the analysis of section 4.1.**
>
> A1: The weight & mask averaging (WMA) is within the framework of Section 4.1 and we have updated the corresponding analysis to make it clearer. Specifically, after collecting the model samples using the random mask, the WMA process can combine the collected samples into an approximation of the posterior distribution, which has been used in Bayesian methods. For example, [R1] used a similar procedure to approximate the mean of the posterior distribution over neural network weights, and achieved good performance.
>
> [R1] Maddox, Wesley J., et al. "A simple baseline for bayesian uncertainty in deep learning." NeurIPS (2019).
>
> **Q2: Figure 5 shows that WMA significantly reduces the ECE value, which is much more obvious than the random mask. The analysis of section 4.2 seems questionable since the major confidence calibration comes from WMA instead of the Bayesian formulation.**
>
> A2: We politely disagree with the reviewer's view. Both random mask (RM) and weight & mask averaging (WMA) are important and indispensable to improving confidence calibration.
>
> (i) Empirically, we have done ablation studies in Section 5.4 to show the effect of RM and WMA. We find that when only WMA is used (CigL w/o RM), the ECE value cannot be effectively reduced. As shown in Figure 5 (b), the blue, green, and pink columns represent RigL, CigL w/o RM, and CigL. The ECE value increases at sparsity 80\% and remains almost constant at sparsity 99\%. For sparsity 90\% and 95\%, it cannot decrease as much as when using RM and WMA together.
>
> We summarize the **changes in ECE values** when using our CigL and when using only the WMA (CigL w/o RM). As shown in the table below, CigL is usually effective in reducing ECE values, but CigL w/o RM either increases ECE values or has only a limited reduction.
> | Sparsity |  80%  |  90%  |  95%  |  99%  |
> | :---------: | -----: | -----: | -----: | -----: |
> | CigL w/o RM (cifar-100, ResNet-50)| 0.0038 | -0.0098 | -0.0144 | -0.0046 |
> | CigL (cifar-100, ResNet-50) | **-0.0010** | **-0.0152** | **-0.0344** | **-0.0269** |
> | CigL w/o RM (cifar-10, Wide-ResNet-22-2) | 0.0060 | 0.0129 | 0.0149 | 0.0017 |
> | CigL (cifar-10, Wide-ResNet-22-2) | **-0.0141** | **-0.0113** | **-0.0104** | **-0.0049**|
>
> (ii) Existing work also supports that WMA does not significantly reduce ECE values. When only using WMA without a random mask, the averaging procedure is equivalent to weight averaging. As mentioned in [R2], although weight averaging can produce better generalization, it does not improve confidence calibration. This suggests that the improvement in confidence calibration comes not only from the WMA but also from the RM.
>
> [R2] Wortsman, Mitchell, et al. "Model soups: averaging weights of multiple fine-tuned models improves accuracy without increasing inference time." ICML, 2022.
>
> **Q3: The process in Algorithm 1 and the Bayesian approximation introduced in section 4.1 are not well aligned, especially when learning the mask. In Algorithm 1, the mask $\textbf{M}$ is updated using weights and gradients. In Section 4.1, the mask $\textbf{M}$ is only related to the weight magnitude.**
>
> A3: Algorithm 1 and Section 4.1 are aligned, which is illustrated as follows. (We have updated the description in the new version)
>
> (i) We can adjust the variational distribution $q(\textbf{M}_l | \textbf{U}_l)$ according to the way we update the mask. In Section 4.1, the variational distribution of mask $\textbf{M}$ expresses how the algorithm selects the important weights, which incorporates the learning of $\textbf{M}$ into the Bayesian paradigm.  When we define $q(\textbf{M}_l | \textbf{U}_l) \propto \exp(\textbf{M}_l  \odot \vert\textbf{U}_l\vert)$, it expresses that weights with large magnitudes are important and are consistent with the pruning step of RigL.
>
> (ii) We can define the variational distribution to encourage larger magnitudes of the weights and gradients so that the learning of $\textbf{M}$ in Algorithm 1 and Section 4.1 will be aligned.
> $q(\textbf{M}_l | \textbf{U}_l) \propto \exp(\textbf{M}_l \odot (\vert\textbf{U}_l\vert + \vert\nabla \textbf{U}_l\vert))$ can be such a definition, which more accurately approximates how Algorithm 1 prunes and regrows the weights. In the pruning step, since the weights have been trained for hundreds of iterations, the gradient magnitudes $\vert\nabla \textbf{U}_l\vert$ can be relatively small compared to the weight magnitudes $\vert\textbf{U}_l\vert$. We can use $q(\textbf{M}_l | \textbf{U}_l) \propto \exp(\textbf{M}_l \odot \vert\textbf{U}_l\vert)$ to approximate the distribution. In the regrowth step, since the inactive weights are all zero, we only need to compare the gradient magnitudes and have $q(\textbf{M}_l | \textbf{U}_l) \propto \exp(\textbf{M}_l \odot \vert\nabla\textbf{U}_l\vert)$.

---

> > ### Author Response · Authors · 2022-11-13
> > **Response to reviewer 3t4S (Q4-Q7)**
> >
> > **Q4: Also, weights $\textbf{W}$ and $\textbf{M}$ are updated iteratively. Why the approximation of section 4.1 still holds when the learning of $M$ is so different?**
> >
> > A4: The iterative update of $\textbf{W}$ and $\textbf{M}$ is a common way in coordinate ascent variational inference when we have multiple groups of parameters [R3, R4]. Therefore, both Algorithm 1 and Section 4.1 can iteratively optimize the parameters and are consistent.
> >
> > [R3] Jordan, Michael I., et al. "An introduction to variational methods for graphical models." Machine learning 37.2 (1999): 183-233.
> >
> > [R4] Wan, Neng, Dapeng Li, and Naira Hovakimyan. "F-divergence variational inference." NeurIPS (2020): 17370-17379.
> >
> > **Q5: The formulation of $q(\textbf{M}_l | \textbf{U}_l)$ seems wired: $\textbf{M}_l$ appeared in both left and right of $\propto$. In addition, it's confusing whether $\textbf{M}_l$ is learned or sampled from a distribution.**
> >
> > A5: (i) We define $q(\textbf{M}_l | \textbf{U}_l)$ to reflect how Algorithm 1 selects the mask for a given weight. Different $\textbf{M}_l$ has different likelihoods. If we think that large weight magnitude is better, then we can define $\exp(\textbf{M}_l \odot \vert\textbf{U}_l\vert)$ to provide a greater likelihood for $\textbf{M}$ if it chooses weights with larger magnitudes.
> >
> > (ii) In Algorithm 1, $\textbf{M}$ is learned through pruning and regrowth steps. In Section 4.1, to align with Algorithm 1, we find an estimate that maximizes $q(\textbf{M}_l | \textbf{U}_l)$, which can be viewed as approximating the distribution $q(\textbf{M}_l | \textbf{U}_l)$ with the estimate. Then, $\textbf{M}$ has 0 or 1 values and is consistent between Algorithm 1 and Section 4.1.
> >
> > **Q6: In the final section, the authors provide four arguments in (a), (b), (c), and (d), and they are not very well supported. For example, in (b), the authors say that "It gradually becomes equivalent to a shallow model." How this conclusion arrives? There are no experiments to show measurements like effective depth.**
> >
> > A6: (i) The four arguments in (a), (b), (c), and (d) are first supported by intuition. In the discussion, we draw analogies between model types such as "shallow models" and "poor models" in terms of model accuracy (expressiveness) and size. Consistent with the previous definition of double descent [R5, R6], we consider sparsity as a measure of model size. Intuitively, as we gradually reduce the model size (increase the sparsity), we will go through four stages, including dense deep model, sparse deep model, shallow model, and poor model.
> >
> > (ii) The four arguments are also supported by our sparse training experiments on ResNet-50 at CIFAR-100. As shown in Figure 1 (c), we can infer the model type by sparsity and accuracy:
> > - For 99.7\% sparsity, the accuracy of the model is 41.7\%, which is similar to a shallow model.
> > - For 99.9\% sparsity, the accuracy of the model is 23.5\%, which can be viewed as a poor model.
> >
> > (iii) A very detailed support of these four arguments is beyond the main scope of this paper. We will investigate this double-drop phenomenon more comprehensively in the next study. In this paper, we hope to give some valuable intuitions and encourage future research.
> >
> > (iv) Thank you for suggesting effective depth as a measurement. We will use it in our future study.
> >
> > [R5] Nakkiran, Preetum, et al. "Deep double descent: Where bigger models and more data hurt." Journal of Statistical Mechanics: Theory and Experiment 2021.12 (2021): 124003.
> >
> > [R6] Somepalli, Gowthami, et al. "Can Neural Nets Learn the Same Model Twice? Investigating Reproducibility and Double Descent from the Decision Boundary Perspective." CVPR. 2022.
> >
> > **Q7: What is ADOPT? This is not explained in the paper.**
> >
> > A7: This is a typo. It is an old name for CigL and we have corrected it in the new version.

---

> ### Author Response · Authors · 2022-11-17
> **Thank you again for your review and hope you may take a quick look at our responses.**
>
> Dear Reviewer,
>
> Thank you very much for taking your valuable time to review our paper. We are fortunate to have received very informative feedback from your comments, which is indispensable for us to polish the paper into a better version.
>
> We were wondering **if you could take a few minutes to take a quick look at our responses** and **let us know if you have any remaining questions or concerns so that we can address them before the deadline**. Much appreciate!
>
> Also, if you feel that your original concerns have been resolved, we would appreciate it if you would update your evaluation to reflect this. Thank you!
>
> Sincerely,
>
> Authors

---

> > ### Comment · Reviewer_3t4S · 2022-11-17
> > **Follow up questions**
> >
> > Dear authors,
> >
> > Thanks for your detailed response. I still have some questions regarding WMA and the analysis of section 4.1. I took a look at [R1], and in [R1], they directly construct the posterior distribution of model weights based on $\theta_{\text{SWA}}$ and its variance. It seems that using WMA or SWA alone cannot be directly connected to Bayesian methods, and you also need weights along the WMA or SWA process. In the proposed method, the authors did not have such constructions. A more direct question is if the authors do not use WMA, then will it affect the analysis of section 4.1? If the analysis of section 4.1 holds for W with or without WMA, then I do not think using WMA can be well connected to the analysis of section 4.1.

---

> > > ### Author Response · Authors · 2022-11-19
> > > **Response to follow-up questions**
> > >
> > > Thank you very much for your reply and valuable comments! Please see our response below to clarify the follow-up questions.
> > >
> > > **Q1: It seems that using WMA or SWA alone cannot be directly connected to Bayesian methods, and you also need weights along the WMA or SWA process.**
> > >
> > > A1: Our method can be seen as approximating the Bayesian posterior with the variational distribution $q(\textbf{W})$ in Equation 3, and WMA can be seen as approximating the mean of the posterior based on samples from $q(\textbf{W})$ using moment matching.
> > >
> > > (i) CigL is connected to the Bayesian approach because of using both deterministic and random masks to explore the weight space. As shown in Equation 3, the design of $\textbf{Z}$ and $\textbf{M}$ results in a hierarchical variational distribution $q(\textbf{W})$, where the hierarchy expands the approximation family and leads to a better posterior approximation capability. In Equation 3, updating the mask is equivalent to updating $\textbf{Z}$ and $\textbf{M}$ to bring $q(\textbf{W})$ closer to the posterior.
> > >
> > > (ii) WMA can be seen as an approximation to the Bayesian integral after we obtain the samples from $q(\textbf{W})$.
> > >
> > > In a similar idea to weight dropout [R2], WMA can be considered as a Bayesian approximation, which is used to approximate the mean of the posterior by moment matching.
> > > - Weight dropout has been shown to be equivalent to the Bayesian approximation method [R2]. After obtaining the final $\textbf{W}$, dropout approximates $\int f(\textbf{W}\textbf{Z}) p(\textbf{Z})d\textbf{Z}$ using $f(E(\textbf{W}\textbf{Z}))$, where $f$ is the neural network and the mean $E(\textbf{W}\textbf{Z})=\int \textbf{W}\textbf{Z} p(\textbf{Z})d\textbf{Z}$ [R3]. This approximation actually approximates the whole posterior by the first moment of the posterior (i.e., the mean).
> > > - For our WMA, since we assume a hierarchy, we collect multiple samples of $\textbf{W}$ and $\textbf{Z}$. Then, the WMA is used to approximate the first moment of the posterior which is used as an approximation of the posterior itself [R3].
> > > - In addition, if we really want the second moment, it is straightforward to obtain an estimation based on samples using moment matching again, similar to [R1]. We do not estimate the second moment since the sparse training typically wants a single sparse model in the end to reduce both computational and memory costs, and we find that using the posterior mean already significantly improves the calibration of the sparse training.
> > >
> > > [R1] Maddox, Wesley J., et al. "A simple baseline for bayesian uncertainty in deep learning." NeurIPS (2019).
> > >
> > > [R2] Gal, Yarin, and Zoubin Ghahramani. "Dropout as a bayesian approximation: Representing model uncertainty in deep learning." ICML 2016.
> > >
> > > [R3] Srivastava, Nitish, et al. "Dropout: a simple way to prevent neural networks from overfitting." JMLR 15.1 (2014): 1929-1958.
> > >
> > > **Q2: If the authors do not use WMA, then will it affect the analysis of section 4.1?"**
> > >
> > > Without the use of WMA, the analysis in Section 4.1 would be a non-hierarchical Bayesian method or a poor approximation to a hierarchical Bayesian approach.
> > >
> > > (i) In the absence of WMA, the algorithm can be viewed as a non-hierarchical variational inference. As described in Section 4.3, using the final $\textbf{W}$ for prediction without WMA is equivalent to using weight dropout in RigL. Thus, the analysis in Section 4.1 will be updated in a similar way to Section 3 in [R2] which shows that weight dropout can be viewed as a non-hierarchical Bayesian approximation.
> > >
> > > (ii) Without WMA, the algorithm can also be viewed as a poor approximation to hierarchical variational inference.
> > > - If we continue to interpret the algorithm without WMA using the current analysis structure from Section 4.1, then how we generate the final posterior approximation will change.
> > > - In this case, although the algorithm is still a Bayesian approximation, we only use the final $\textbf{W}$ to represent the posterior, which does not effectively capture all the information we explore from the weight space and the increased correlation between $\textbf{Z}$ and $\textbf{W}$.
> > > - Therefore, it turns out to be a bad hierarchical approximation, which limits its power.
> > >
> > > (iii) Empirically, we also find that using only the final $\textbf{W}$ without WMA (weight dropout) does not work well. As shown in Tables 2 and 3 in Section 5.2, "RigL+Weight dropout" typically has lower accuracy and higher ECE compared to our CigL.
> > >
> > > [R2] Gal, Yarin, and Zoubin Ghahramani. "Dropout as a bayesian approximation: Representing model uncertainty in deep learning." ICML 2016.

---

> > > > ### Comment · Reviewer_3t4S · 2022-11-28
> > > > **Thanks for the response**
> > > >
> > > > Dear authors,
> > > >
> > > > Thanks for your response. The previous response and this response clarify most of my concerns. I will increase my rating.

---

> > > > > ### Author Response · Authors · 2022-11-28
> > > > > **Thank you again!**
> > > > >
> > > > > Dear Reviewer,
> > > > >
> > > > > Thank you again for your valuable comments and questions!

---

### Author Response · Authors · 2022-11-13
**Common Response to all Reviewers**

We thank all reviewers for their insightful comments. The main contributions of this work as stated by the reviewers are as follows:

- The first identification and study of the problem of calibration in sparse training is important and is of great interest to the sparse training research community **[Reviewer JN9S, db8U]**
- The proposed new sparse training method with deterministic mask & random mask is interesting and novel **[Reviewer 3t4S, HnVS, db8U]**
- The connection to Gaussian processes and hierarchical variational inference is provided as a theoretical justification **[Reviewer 3t4S, JN9S, db8U]**
- The connection with dropout is explicitly identified and discussed **[Reviewer db8U]**
- Solid experiments and clear visualization; ablation of each component **[Reviewer 3t4S, HnVS, db8U]**
- The paper is well written, clearly motivated, and of high quality **[Reviewer HnVS, db8U]**
- Reproducible (submitted code with running instructions) **[Reviewer 3t4S]**

We would like to emphasize the significant contribution of this paper to sparse training. While being applied in a wide variety of applications, sparse training’s reliability remains unstudied. This work provides the first investigation of the calibration in sparse training, paving the way toward applying sparse training in real-world decision making systems. In the paper, we reveal the issue of severe uncalibrated prediction in existing sparse training methods, and propose a simple yet effective algorithm to greatly improve the calibration while still keeping the high accuracy. We have provided both theoretical support and extensive experimental results over different datasets and architectures to demonstrate the effectiveness of our method. Our method only incurs a small overhead and can be used as a drop-in replacement for existing sparse training methods.  We believe this work fills an important gap in sparse training and hope the reviewers can consider our response in their final evaluation.

---

### Author Response · Authors · 2022-11-13
**Revision Summary**

We thank all reviewers for their constructive review and have revised the paper accordingly. The main changes are the following (highlighted in blue):

- Section 1. We revise our first contribution to avoid over-claim.
- Section 1. We add a description of the ECE metric to avoid confusion.
- Section 2.1 & Appendix C.5. We add an additional discussion about weight exploration in sparse training.
- Section 2.2. We add an additional discussion about reliability in sparse models.
- Section 3. We revise Algorithm 1 to avoid confusion.
- Section 4. We exchange Sections 4.1 and 4.2 to clarify the theory's motivation.
- Section 4.2. We revise the description to be more aligned with Algorithm 1.
- Section 6. We revise the description of the 4 phases to avoid confusion.
- Appendix C.2. We add additional experiments on SET to study the reliability of sparse training.
- Appendix C.3. We add an additional comparison between our CigL and Sup-tickets.
- Appendix C.4. We add additional experiments to test the effect of the weight \& mask averaging process.

---

### Author Response · Authors · 2022-11-16
**Please kindly let us know if you have any questions, thank you!**

We are extremely grateful for the helpful comments and constructive feedback on our paper. Please kindly let us know if you have any remaining questions or concerns so that we can address them before the deadline. Thank you! Much appreciate!

Alternatively, if you feel that your original concerns have been addressed, we would appreciate it if you could update your evaluation to reflect this. Thank you!

---

### Decision · Program_Chairs · 2023-01-20

**Decision:**

Accept: poster

**Justification For Why Not Higher Score:**

This paper is a nice productive result, but it's still quite niche. It addresses one specific shortcoming (calibration) of one specific algorithm (RigL, and potentially SET based on additional results) in a specific part of the pruning literature (sparse training). It's a useful observation and improvement in that part of the space, but it's a niche community who will be interested int his work and it's unlikely to have general implications for multiple parts of the machine learning world. In short, it's not significant enough to deserve more than a poster.

**Justification For Why Not Lower Score:**

It's a good technical result that improves on a popular baseline in the pruning world. It should be accepted.

**Metareview: Summary, Strengths And Weaknesses:**

**Summary:** This paper studied the problem of poor calibration in sparse neural networks. Sparse networks (such as those produced by sparse training methods and pruning) are more over-confident than unpruned (dense) neural networks. This paper addresses this problem in a specific setting, RigL. It does so in a clever way, combining a low-sparsity random pruning mask on each step (essentially the drop-connect variant of dropout) with the pruning mask learned by RigL. This results in improved calibration without sacrificing the accuracy of networks produced by RigL

**Strengths:** This is a nice win: it's a good observation to look at calibration and the proposed solution is (i) simple and (ii) effective. It's a solid technical result.

**Weaknesses:**
* This method sacrifices one of the beneficial aspects of RigL: that training can happen in a truly sparse way outside of the points in training where the mask changes.
* A bit of overclaiming here and there in the paper, which appears to have been somewhat cleaned up during the review and discussion process. I would still advise the authors to further scale back their claims where possible. You made it through the review process - there's no need to exaggerate from here. At this point, the paper will live or die by the quality of the technical work, and it will only turn people off to overclaim.

**Note From Pc:**

if the above contains the word "oral" or "spotlight" please see: "oral" presentation means -> notable-top-5% and "spotlight" means -> notable-top-25%. As stated in our emails, we are disassociating presentation type from AC recommendations

**Summary Of Ac-Reviewer Meeting:**

N/A